# *Rasgrp1* mutation increases naïve T-cell CD44 expression and drives mTOR-dependent accumulation of Helios+ T cells and autoantibodies

Stephen R Daley[1†], Kristen M Coakley[2†], Daniel Y Hu[1†], Katrina L Randall[1,3†], Craig N Jenne[4†], Andre Limnander[2†], Darienne R Myers[2], Noelle K Polakos[2], Anselm Enders[1], Carla Roots[1], Bhavani Balakishnan[5], Lisa A Miosge[1], Geoff Sjollema[5], Edward M Bertram[1,5], Matthew A Field[1], Yunli Shao[1], T Daniel Andrews[1], Belinda Whittle[5], S Whitney Barnes[6], John R Walker[6], Jason G Cyster[7], Christopher C Goodnow[1,5*‡], Jeroen P Roose[2*‡]

[1]Department of Immunology, John Curtin School of Medical Research, The Australian National University, Canberra, Australia; [2]Department of Anatomy, University of California, San Francisco, San Francisco, United States; [3]Department of Immunology, Canberra Hospital and ANU Medical School, The Australian National University, Canberra, Australia; [4]Institute of Infection, Immunity and Inflammation, University of Calgary, Calgary, Canada; [5]Australian Phenomics Facility, John Curtin School of Medical Research, The Australian National University, Canberra, Australia; [6]Department of Genetics, Genomics Institute, Novartis Research Foundation, San Diego, United States; [7]Department of Microbiology and Immunology, Howard Hughes Medical Institute, University of California, San Francisco, San Francisco, United States

**\*For correspondence:** Chris. Goodnow@anu.edu.au (CCG); jeroen.roose@ucsf.edu (JPR)

†These authors contributed equally to this work

‡These authors also contributed equally to this work

**Competing interests:** The authors declare that no competing interests exist.

**Reviewing editor**: Shimon Sakaguchi, Osaka University, Japan

**Abstract** Missense variants are a major source of human genetic variation. Here we analyze a new mouse missense variant, *Rasgrp1Anaef*, with an ENU-mutated EF hand in the Rasgrp1 Ras guanine nucleotide exchange factor. *Rasgrp1Anaef* mice exhibit anti-nuclear autoantibodies and gradually accumulate a CD44hi Helios+ PD-1+ CD4+ T cell population that is dependent on B cells. Despite reduced Rasgrp1-Ras-ERK activation in vitro, thymocyte selection in *Rasgrp1Anaef* is mostly normal in vivo, although CD44 is overexpressed on naïve thymocytes and T cells in a T-cell-autonomous manner. We identify CD44 expression as a sensitive reporter of tonic mTOR-S6 kinase signaling through a novel mouse strain, *chino*, with a reduction-of-function mutation in *Mtor*. Elevated tonic mTOR-S6 signaling occurs in *Rasgrp1Anaef* naïve CD4+ T cells. CD44 expression, CD4+ T cell subset ratios and serum autoantibodies all returned to normal in *Rasgrp1AnaefMtorchino* double-mutant mice, demonstrating that increased mTOR activity is essential for the *Rasgrp1Anaef* T cell dysregulation.

## Introduction

Positive and negative selection of thymocytes generates a population of T lymphocytes with a broad spectrum of antigen-specific T cell receptors (TCR) (*Starr et al., 2003*; *Kortum et al., 2013*). It was recognized early on that the small GTPase Ras plays a role (*Swan et al., 1995*). Three Ras guanine exchange factor (RasGEF) families can activate Ras: SOS, RasGRP, and RasGRF (*Stone, 2011*). Following TCR engagement, *Son of Sevenless* (SOS)-1 and -2 are recruited to the plasma membrane via a Grb2-phospho-LAT interaction. Simultaneously, the second messenger diacylglycerol (DAG), generated via PLCγ, directly recruits Ras guanine nucleotide releasing protein 1 (Rasgrp1) to the plasma membrane

**eLife digest** Our DNA contains more than three billion nucleotides. Each of these nucleotides can be an A, C, G or T, and groups of three neighboring nucleotides within our DNA are used to represent the 20 amino acids that are used to make proteins. This means that changing just one nucleotide can cause one amino acid to be replaced by a different amino acid in the protein encoded by that stretch of DNA: AAA and AAG code for the amino acid lysine, for example, but AAC and AAT code for asparagine. Known as missense gene variants, these changes can also increase or decrease the expression of the gene.

Every person has thousands of missense gene variants, including about 12,000 inherited from their parents. Sometimes these variants have no consequence, but they can be harmful if replacing the correct amino acid with a different amino acid prevents the protein from performing an important task. In particular, missense gene variants in genes that encode immune system proteins are likely to play a role in diseases of the immune system. For example, variants near a gene called Rasgrp1 have been linked to two autoimmune diseases – type 1 diabetes and Graves' disease—in which the immune system mistakenly attacks the body's own cells and tissues.

Now Daley et al. have shed new light on the mechanism by which a missense gene variant in Rasgrp1 can cause autoimmune diseases. Mice with this mutation show signs of autoimmune disease, but their T cells—white blood cells that have a central role in the immune system – develop normally despite this mutation. Instead, Daley et al. found that a specific type of T cell, called T helper cells, accumulated in large numbers in the mutant mice and stimulated cells of a third type—immune cells called B cells—to produce autoantibodies. The production of autoantibodies is a common feature of autoimmune diseases.

Daley et al. traced the origins of the T helper cells to excessive activity on a signaling pathway that involves a protein called mTOR, and went on to show that treatment with the drug rapamycin counteracted the accumulation of the T helper cells and reduced the level of autoimmune activity. In addition to exemplifying how changing just one amino acid change can have a profound effect, the work of Daley et al. is an attractive model for exploring how missense gene variants in people can contribute to autoimmune diseases.

(*Ebinu et al., 1998*). Biochemically, Rasgrp1 and SOS1 synergize to induce high-level Ras activation (*Roose et al., 2007*) and Rasgrp1 serves a critical role in priming SOS1 via Rasgrp1-produced RasGTP (*Das et al., 2009*). Consequentially, thymocyte development is severely impaired in *Rasgrp1*-deficient mice (*Dower et al., 2000*), and not compensated for by SOS RasGEFs. Additionally, there is only minimal compensation for loss of Rasgrp1 coming from Rasgrp3 or Rasgrp4 (*Zhu et al., 2012*; *Golec et al., 2013*). *Rasgrp1*-deficient mice exhibit a strong defect in positive selection and impaired ERK phosphorylation in thymocytes (*Dower et al., 2000*; *Priatel et al., 2002*). The importance of the canonical Rasgrp1-RasGTP-RAF-MEK-ERK pathway for developing thymocytes is further underscored by impaired positive selection in ERK-1 and -2 doubly deficient mice (*Fischer et al., 2005*).

Although Rasgrp1 plays a critical role in the activation of Ras, relatively little is known about its regulation in T lymphocytes or the in vivo importance of such regulation. In addition to membrane recruitment via its DAG-binding C1 domain (*Ebinu et al., 1998*), Rasgrp1's GEF activity is enhanced by inducible phosphorylation of threonine 184 (*Roose et al., 2005*; *Zheng et al., 2005*). Phospholipase C γ (PLCγ) not only generates DAG but also inositol 1,4,5-trisphosphate (IP3), which binds to IP3 receptors on the endoplasmic reticulum to activate the calcium pathway (*Feske, 2007*). Interestingly, Rasgrp1 also contains a pair of EF hands, motifs that often bind calcium, which induces conformational changes (*Gifford et al., 2007*). Rasgrp1 has been reported to bind calcium in vitro (*Ebinu et al., 1998*). In chicken DT40 B cells, the first $EF_1$ domain enables the recruitment function of a C-terminal PT domain (plasma membrane targeting domain) that cooperates with the C1 domain to recruit Rasgrp1 to the membrane (*Tazmini et al., 2009*). Notably, the PT domain contribution is substantial in BCR-stimulated B cell lines, very modest in T cell lines, and negligible in fibroblasts (*Beaulieu et al., 2007*). Genetic deletion of Rasgrp1's 200 C-terminal amino acids reduces the formation of mature thymocytes in *Rasgrp1*$^{d/d}$ mice (*Fuller et al., 2012*). Our recent structural studies revealed that Rasgrp1's C terminus contains a coiled-coil dimerization domain (*Iwig et al., 2013*). Rasgrp1 dimerization plays

an important role in controlling Rasgrp1's activity; the second EF hand of one Rasgrp1 molecule packs against the C1 domain of a second molecule in a manner that is incompatible with DAG-binding whereas calcium binding to the first EF hand is predicted to unlock this autoinhibitory dimer interface (*Iwig et al., 2013*). Lastly, it is unknown if Rasgrp1 may signal to pathways other than the canonical Rasgrp1-Ras-RAF-MEK-ERK cascade, although a link between Rasgrp1 and mTOR (mechanistic target of rapamycin) signaling has been proposed (*Gorentla et al., 2011*).

Older *Rasgrp1*-deficient (*Coughlin et al., 2005*) and *Rasgrp1$^{d/d}$* mice (*Fuller et al., 2012*) develop splenomegaly and autoantibodies. In these mouse models, the complete deletion or truncation of Rasgrp1 greatly decreases T cell development in the thymus (*Dower et al., 2000*; *Fuller et al., 2012*), resulting in peripheral T cell lymphopenia followed by accumulation of CD44$^{hi}$ CD62L$^{lo}$ CD4$^+$ T cells (*Priatel et al., 2007*; *Fuller et al., 2012*). Autoimmune phenotypes caused by these mutations have been attributed to compromised T cell selection in the thymus and compensatory expansion of peripheral T cells in response to lymphopenia and/or chronic infection. Hypomorphic missense alleles of the signaling molecules ZAP-70 and LAT also impair T cell development in the thymus and culminate in severe peripheral immune dysregulation. For example, an SKG allele of the kinase ZAP-70 has reduced binding-affinity for phospho-TCRζ and leads to autoimmune arthritis in mice (*Sakaguchi et al., 2003*). Point mutations in ZAP70's catalytic domain that reduce kinase activity to intermediate levels diminish thymic deletion and Foxp3$^+$ Treg differentiation but preserve peripheral T cell activation, resulting in autoantibody formation and hyper-IgE production (*Siggs et al., 2007*). Mutation of a single tyrosine in LAT (LAT$^{Y136F}$) results in hyperproliferative lymphocytes of a T$_H$2 type (*Aguado et al., 2002*; *Sommers et al., 2002*). In each of these cases, peripheral T cell dysregulation is tied to, and potentially explained by, profound deficits in thymic T cell formation.

Single nucleotide variants that cause amino acid substitutions (missense variants; SNVs) or modify the level of gene expression rather than knocking out protein expression are a major form of human genetic variation: most people inherit ~12,000 missense gene variants (*The 1000 Genomes Project Consortium, 2010*). Given the emerging examples of missense alleles having very different immunological consequences from null alleles, mouse models that analyze the consequences of missense variants in key immune genes are needed to understand the pathogenesis of complex human immune diseases. Common tag SNVs near *RASGRP1* are associated with susceptibility to autoimmune (Type 1) diabetes and to thyroid autoantibodies in Graves' disease (*Qu et al., 2009*; *Plagnol et al., 2011*), while 13 unstudied *RASGRP1* missense SNVs are currently listed in public databases. A fruitful approach for identifying missense gene variants that dysregulate immune function has been through *N*-ethyl-*N*-nitrosourea (ENU) mutagenesis (*Nelms and Goodnow, 2001*). Here we describe the analysis of a novel ENU-induced missense variant, *Rasgrp1$^{Anaef}$* that reveals an important in vivo regulatory function of Rasgrp1's EF hands. *Rasgrp1$^{Anaef}$* is distinct from previously described autoimmune mutations in *Rasgrp1*, *Zap70* or *Lat*, as *Rasgrp1$^{Anaef}$* has no detectable effect on thymocyte development in mice with normal TCR repertoires, but results in peripheral accumulation of a distinct population of Helios$^+$ PD-1$^+$ T-helper cells and production of anti-nuclear autoantibodies. In contrast to *Rasgrp1* deletion, the *Rasgrp1$^{Anaef}$* missense variant increases tonic mTOR signaling in naïve CD4$^+$ T cells. Genetic reduction of mTOR function in *Rasgrp1$^{Anaef}$* mice normalizes CD44 expression on naïve CD4$^+$ T cells and abolishes excessive accumulation of effector T cells and autoantibodies, demonstrating a central role for increased mTOR activity in driving immune dysregulation in *Rasgrp1$^{Anaef}$* mice.

## Results

### Identification of the *Rasgrp1$^{Anaef}$* mouse strain with a mutated EF hand in Rasgrp1

As part of a mouse genome-wide screen for immune phenotypes induced by ENU mutagenesis (*Nelms and Goodnow, 2001*), we identified a variant C57BL/6 (B6) pedigree displaying elevated frequencies of CD44$^{hi}$ CD4$^+$ cells (*Figure 1A*), elevated CD44 expression on naïve FOXP3$^-$ CD44$^{lo}$ CD4$^+$ cells (*Figure 1B*) and antinuclear antibodies (ANAs) staining with a homogeneous nuclear pattern (*Figure 1C,D*). The elevated frequency of CD44$^{hi}$ cells trait, which occurred at a frequency consistent with inheritance of a recessive gene variant (*Figure 1A*), was used to map the mutation in an F2 intercross to an interval between 114 and 121.2 Mb on chromosome 2 (*Figure 1—figure supplement 1A*).

Sequencing of the exons of *Rasgrp1*, the only gene within this interval with a known immune function, identified an A to G missense mutation in codon 519 within exon 13 (*Figure 2A*). Whole-exome capture,

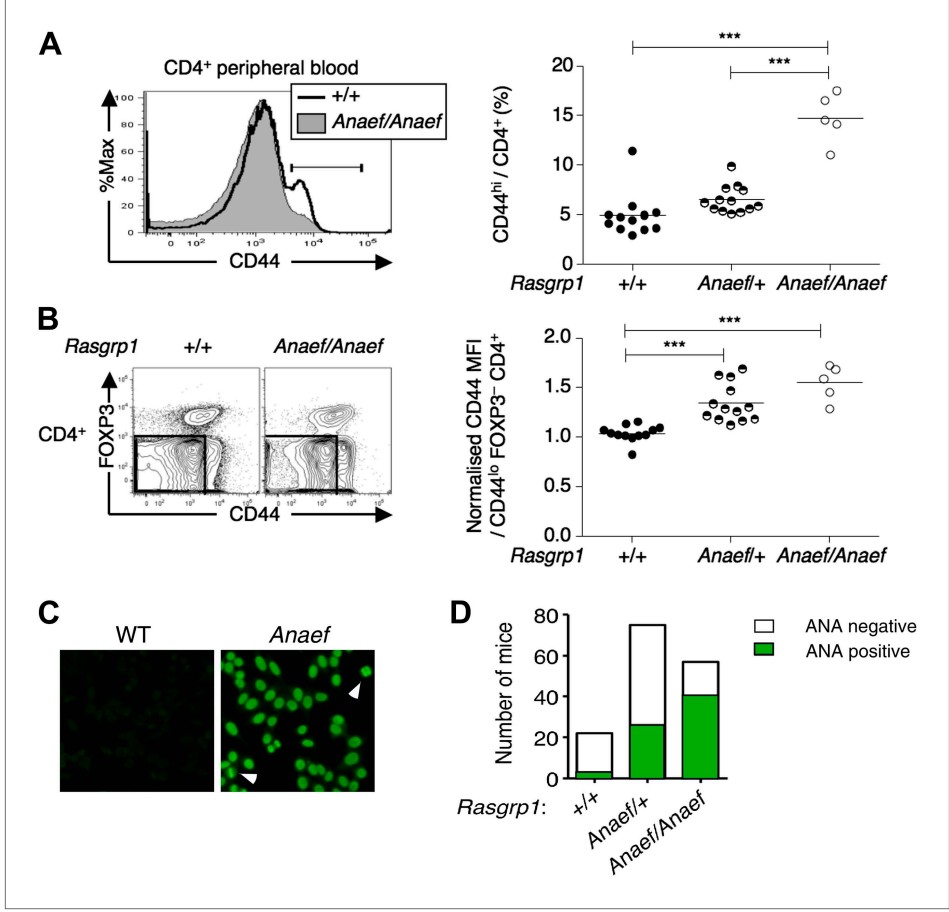

**Figure 1**. An ENU mouse mutant with increased CD44hi CD4 cells and anti-nuclear autoantibodies. (**A** and **B**) Representative flow cytometry showing on peripheral blood CD4$^+$ cells (**A**) CD44 expression with the gate used to define CD44$^{hi}$ cells and (**B**) FOXP3 vs CD44 phenotype including normalized CD44 Mean Fluorescence Intensity (MFI) of the gated CD44$^{lo}$ FOXP3$^-$ subset from *Rasgrp1*$^{+/+}$ (WT), heterozygous *Rasgrp1*$^{Anaef/+}$ or homozygous *Rasgrp1*$^{Anaef/Anaef}$ mice. Statistical analysis (right) used unpaired Student's t tests where each symbol represents an individual mouse; ***p<0.001. (**C**) Antinuclear antibodies (ANA) in diluted blood plasma from a B6xB10. *Rasgrp1*$^{Anaef}$ mouse and wildtype littermate, measured by indirect immunofluorescence on HEp-2 cells. Note homogeneous nuclear staining of interphase cells and positive chromatin bars in dividing cells (marked with arrow). Magnification 20 ×. (**D**) Quantitation of positive ANA results for wildtype, *Rasgrp1*$^{Anaef/+}$ and *Rasgrp1*$^{Anaef/Anaef}$ C57Bl/6xC57Bl/10 siblings tested at 15 weeks of age.

The following figure supplements are available for figure 1:

**Figure supplement 1**. Mapping and genotyping of the Rasgrp1 mutation in ENU mutant mice with anti-nuclear antibodies and CD44hi phenotype.

sequencing and computational analysis of DNA from an affected mouse (*Andrews et al., 2012*) identified this mutation as the only novel single-nucleotide variant within the interval of interest on chromosome 2 (data not shown). The mutant codon, located in Rasgrp1's second EF hand (EF2), encodes a neutral amino acid glycine (G) instead of the normal arginine (R), a large polar molecule with a positive charge (Rasgrp1 R519G; *Figure 2A,B*). Rasgrp1's arginine residue at 519 is also found in Rasgrp-2, and -4, and in EF3 of calcium and integrin binding protein (CIB) (*Figure 2B*). EF hands typically come in pairs separated by a linker and calcium binding subsequently alters the angle between helices E and F in proteins such as calmodulin (CaM) (*Figure 2C*) (*Grabarek, 2006*; *Gifford et al., 2007*). Unique to Rasgrp1, this linker is unusually short. Furthermore, biophysical studies revealed that Rasgrp1's EF2 does not bind calcium, that the E helix is non-existent in EF2, but instead has evolved as a critical loop forming an autoinhibitory interface with the C1 domain (*Figure 2D,E*) (*Iwig et al., 2013*).

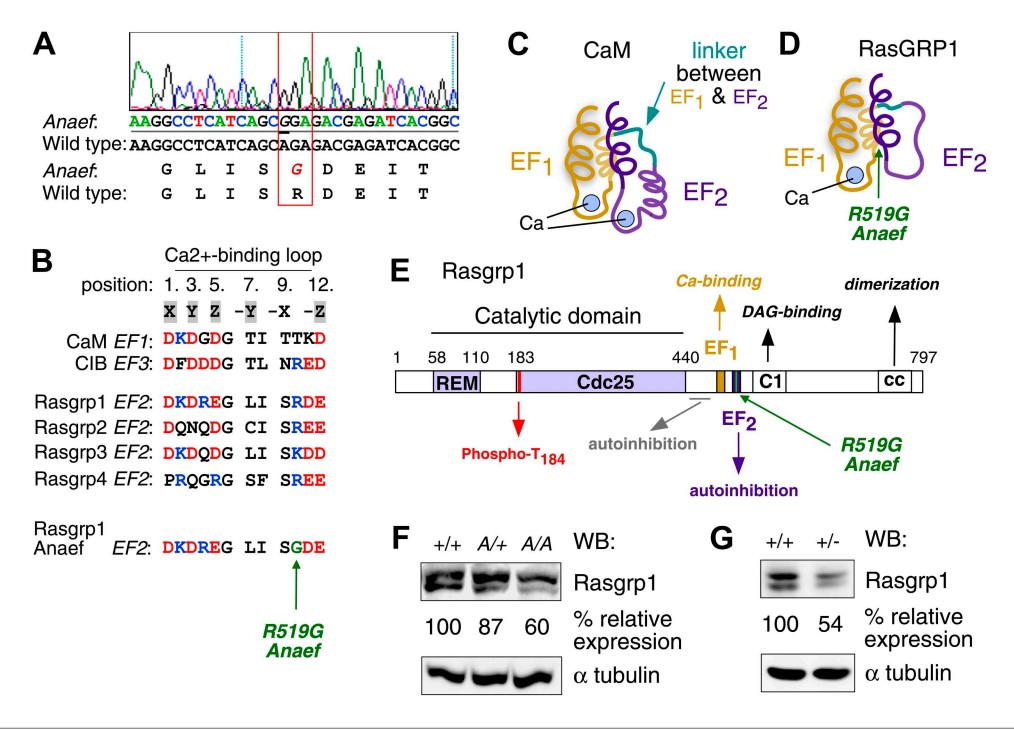

**Figure 2**. Mapping of the ENU mutation to the autoinhibitory second EF hand domain in Rasgrp1. (**A**) Sequence trace of *Rasgrp1^Anaef* exon 13 aligned to wildtype Rasgrp1 sequence. (**B**) Sequence comparison of CaM (calmodulin), CIB (calcium and integrin binding protein) and Rasgrp1, -2, -3 and -4 EF hands with conserved residues highlighted. (Acidic, red; Basic, blue; Calcium-binding residues, highlighted in grey; ENU-mutated residue in green). (**C and D**) Model of a typical pair of EF hands with two calcium (Ca)-binding loops each flanked by N- and C-terminal α-helices based on CaM (**C**). Model including the atypical second EF hand in RasGRP1 (**D**). RasGRP1's second EF hand does not bind calcium and the E helix has evolved into an autoinhibitory domain (***Iwig et al., 2013***). (**E**) Linear schematic of Rasgrp1 protein domains, phosphorylation site threonine 184, and position of R519G mutation in the second EF hand that evolved into a domain for autoinhibition. (**F and G**) Western blot for Rasgrp1 protein in thymocytes from *Rasgrp1^Anaef/Anaef* (A/A), heterozygous *Rasgrp1^Anaef/WT* (A/+), heterozygous *Rasgrp1^Null/WT* (+/−), and wildtype (+/+) mice. Blot was reprobed for α tubulin as a loading control. Relative Rasgrp1 expression was calculated and is shown. Note the expression of Rasgrp1's typical doublet, thought to be due to alternative translation initiation (***Poon and Stone, 2009***). Representative blots of at least three independent experiments.

The ENU-generated allele was named '*Rasgrp1^Anaef*' to reflect the combination of antinuclear antibody (ANA) production and the amino acid substitution in the EF hand. Genotyping of this mutation in multiple generations of B6 offspring (***Figure 1—figure supplement 1B,C***) demonstrated that inheritance of the *Rasgrp1^Anaef* allele was well correlated with the immunological abnormalities described above and below. ANAs were present in 70% of homozygous *Rasgrp1^Anaef/Anaef* mice and 35% of heterozygous *Rasgrp1^Anaef/+* mice (***Figure 1D***), compared to 5% of wildtype B6 mice, indicating a gene dosage effect. The R519G substitution caused an approximate 40% decrease in Rasgrp1 protein levels in homozygous *Rasgrp1^Anaef/Anaef* thymocytes (***Figure 2F***). Since heterozygous *Rasgrp1^+/−* thymocytes express half the Rasgrp1 dosage (***Figure 2G***) but do not display an abnormal immune phenotype (***Dower et al., 2000***), whereas heterozygous *Rasgrp1^Anaef/+* mice do exhibit abnormal immune phenotypes (***Figure 1B,D***), we conclude that the immune dysregulation in mice bearing the *Rasgrp1^Anaef* allele is caused by the specific R519G alteration and not simply by a reduction of Rasgrp1 protein levels. In the remainder of the manuscript we discuss the analysis of homozygous *Rasgrp1^Anaef/Anaef* mice and refer to these as *Rasgrp1^Anaef* mice.

### *Rasgrp1^Anaef* preserves Rasgrp1 function for T cell selection in the thymus

Analysis of *Rasgrp1^Anaef* mice revealed a striking contrast to the published *Rasgrp1*-deficient and *Rasgrp1^d/d* mouse models, which have T cell developmental defects that result in low thymic T cell

output (*Dower et al., 2000*; *Priatel et al., 2007*; *Fuller et al., 2012*). The frequency and number of mature CD4+ and CD8+ single positive (SP) thymocyte subsets was normal in *Rasgrp1^Anaef* mice, whereas these subsets were markedly decreased in *Rasgrp1^-/-* (knockout) mice analyzed in parallel (*Figure 3A,B*). Unlike the knockout allele, the *Rasgrp1^Anaef* mutation did not decrease the frequency of CD69^hi TCRβ^hi cells amongst CD4+CD8+ double positive (DP) thymocytes (*Figure 3C*) or the number of Foxp3+ CD4SP cells in the thymus (*Figure 3B*). Even in bone marrow chimeras reconstituted with a mixture of CD45.2+

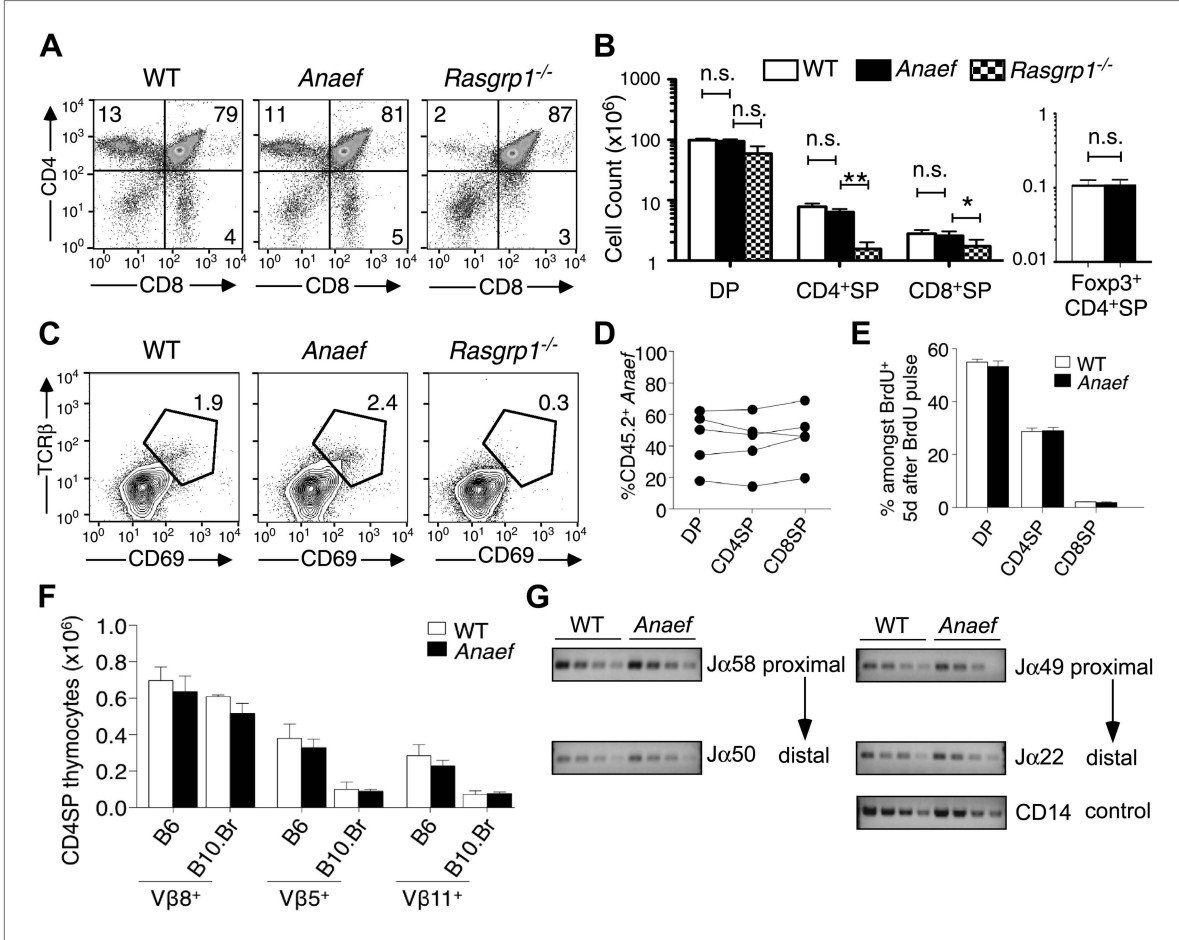

**Figure 3**. *Rasgrp1^Anaef* differs from *Rasgrp1^null* by preserving T cell selection in the thymus. (**A**) CD4/CD8 phenotype of thymocytes in wildtype, *Rasgrp1^Anaef*, and *Rasgrp1^-/-* mice. (**B**) Number of DP, CD4SP and CD8SP as well as Foxp3+ CD4SP thymocytes in wildtype (n = 36), *Rasgrp1^Anaef* (n = 35), and *Rasgrp1^-/-* (n = 3) mice. Student's t tests p value symbols: **p<0.005, *p<0.05. (**C**) Percentages of CD69^hi TCRβ^hi cells among DP cells. Representative of three mice per genotype. (**D**) Competitive repopulation of thymic subsets in irradiated CD45.1+ B6 recipient mice reconstituted with a mixture of CD45.1+ WT and CD45.2+ *Rasgrp1^Anaef* bone marrow cells. The percentage of CD45.2+ cells amongst the indicated thymocyte subsets was determined 8 weeks after reconstitution. Lines connect measurements from individual mice. Data are representative of three separate chimera cohorts. (**E**) Mixed bone marrow chimeras bearing CD45.1+ WT and either CD45.2+ WT or CD45.2+ *Rasgrp1^Anaef* hematopoietic cells were injected i.p. with 1 mg BrdU 5 days before analysis. The percentages of DP, CD4SP and CD8SP cells amongst BrdU-labeled CD45.2+ *Rasgrp1^Anaef* (black columns) or CD45.2+ WT (white columns) thymocytes were determined and the mean ± SEM (n = 5 mice per group) is shown. See *Figure 3—figure supplement 1A* for flow cytometry gates. (**F**) Mean ± SEM number of CD4SP thymocytes expressing TCR Vβ8, which is not superantigen reactive, or Vβ5 or Vβ11 that recognize endogenous retroviral superantigen presented by the I-E MHCII molecule, comparing B6 mice lacking I-E and B10.BR expressing the I-E (n = 3 mice per group compiled from two experiments). (**G**) Analysis of Jα usage in sorted CD4SP thymocytes of wildtype or *Rasgrp1^Anaef* mice following published methods. PCR was performed using serial 1:2 dilutions of template DNA with primer sets that reveal rearrangements proximal (Jα58, Jα49) or distal (Jα50, Jα22) to the two promoters. *Figure 2G* is representative of three independent experiments. See *Figure 3—figure supplement 1* for details on promoter use, additional Jα segments, and published methods.

The following figure supplements are available for figure 3:

**Figure supplement 1**. Comparison of thymocyte maturation kinetics in WT and *Rasgrp1^Anaef*.

*Rasgrp1^Anaef* and CD45.1^+ wildtype marrow, the *Rasgrp1^Anaef* thymocytes exhibited no competitive disadvantage as they matured from DP to SP cells (***Figure 3D***). Injection into mice of 5-bromo-2′-deoxyuridine (BrdU) to pulse-label a cohort of proliferating DP thymocytes followed by analysis on day 5 demonstrated that the kinetics of maturation into SP cells was unaffected by the *Rasgrp1^Anaef* mutation (***Figure 3E***, ***Figure 3—figure supplement 1A***). There was also normal deletion of Vβ5^+ and Vβ11^+ SP thymocytes upon self-superantigen/I-E^k recognition in B10.Br mice (***Figure 3F***) and similar usage of TCRα Jα segments in wildtype and *Rasgrp1^Anaef* CD4SP thymocyte populations (***Figure 3G***, ***Figure 3—figure supplement 1B***). Thus, analysis of the thymus of *Rasgrp1^Anaef* mice with a diverse TCR repertoire revealed no abnormalities in positive selection, Foxp3^+ T-regulatory (T-reg) cell differentiation or clonal deletion, in striking contrast to previously described *Rasgrp1* mutations.

## Anaef diminishes canonical Rasgrp1-Ras-ERK signaling in response to in vitro stimulation

Despite the normal thymic development in *Rasgrp1^Anaef* animals, there was a striking biochemical effect of the *Rasgrp1^Anaef* mutation on activation of the canonical Rasgrp1-Ras-ERK signaling pathway in a range of in vitro stimulation assays. GFP-tagged wildtype- or *Anaef*- Rasgrp1 was transiently expressed in RasGRP1-deficient Jurkat cells (JPRM441) (***Roose et al., 2005***), which were either left unstimulated or stimulated with a combination of PMA (a synthetic analog of diacylglycerol) and ionomycin (a calcium ionophore). Gating on cells with different GFP intensities (***Figure 4—figure supplement 1A***) revealed that Rasgrp1^Anaef was hypomorphic (partial loss of function) for activating the Ras-ERK pathway: in GFP^+ cells expressing Rasgrp1^Anaef there was only low ERK phosphorylation (P-ERK) and this was only modestly increased when PMA and ionomycin were added (***Figure 4A***). By contrast, GFP^+ cells expressing high levels of wildtype Rasgrp1 vector induced 5-times higher P-ERK spontaneously and this was doubled by PMA and ionomycin stimulation. Next, we stably reconstituted the JPRM441 cell line, which expresses ~10% of residual wildtype RasGRP1 protein (***Roose et al., 2005***) with *Rasgrp1^Anaef* or *Rasgrp1^wildtype* vectors and selected clones with Rasgrp1 expression levels similar to the parental Jurkat cell line (***Figure 4—figure supplement 1B***). Since JPRM441 cells do not express surface TCR (***Roose et al., 2005***), clonal cell lines were stimulated with PMA followed by RasGTP pull-down assays, which demonstrated that *Rasgrp1^Anaef* decreased PMA-induced GTP-loading of Ras to levels below that of the nontransfected JPRM441 cells (***Figure 4B***). In the same transfected cell lines, PMA-induced P-ERK responses were decreased in *Rasgrp1^Anaef* expressing cells, most notable with the lower dose of PMA (PMA MED; 5 ng/ml) and contrasted the effective induction of P-ERK signals in Jurkat and JPRM441-WT-Rasgrp1 cells (***Figure 4C,D***). Similarly, *Rasgrp1^Anaef* expressing cells demonstrated less potent synergy in P-ERK levels when ionomycin was combined with a very low PMA stimulus (2 ng/ml) (***Figure 4E***, ***Figure 4—figure supplement 1C***). In fact, P-ERK responses in JPRM441-Rasgrp1^Anaef cells were more impaired than in the parental JPRM441 cells, indicating a dominant negative effect, which was also observed at the level of Ras activation (***Figure 4B***). We previously reported a dominant negative effect for ΔDAG-Rasgrp1, a form of Rasgrp1 lacking the DAG-binding C1 domain (ΔDAG) and we postulated that there may be competition with the residual ~10% of wildtype RasGRP1 (***Roose et al., 2005***). As Rasgrp1 is regulated by DAG-driven membrane recruitment (***Ebinu et al., 1998***; ***Roose et al., 2005***) we examined this process for Rasgrp1^Anaef. EGFP-tagged wildtype or *Anaef* Rasgrp1-transfected JPRM441 cells were FACS sorted on low GFP expression to avoid overexpression artifacts and cells were allowed to adhere to coated slides. PMA stimulation resulted in membrane recruitment and cytoplasmic clearing of wildtype Rasgrp1; whereas these events were decreased for Rasgrp1^Anaef (***Figure 4F***).

To test TCR-induced Ras-ERK signaling in thymocytes, we first stimulated thymocytes from wildtype or *Rasgrp1^Anaef* mice with anti-CD3 crosslinking antibodies and probed lysates for tyrosine-phosphorylated proteins to examine the global biochemical effects of the *Rasgrp1^Anaef* mutation. Both thymocyte populations demonstrated similar induction of total phospho-tyrosine patterns and similar activating phosphorylation of Lck and Zap-70 that lie upstream of Rasgrp1 (***Figure 5—figure supplement 1A***). Rasgrp1's GEF activity is also enhanced by phosphorylation on T184 (***Roose et al., 2005***; ***Zheng et al., 2005***). Using a new monoclonal antibody specific for P-T$_{184}$-Rasgrp1 (***Figure 5—figure supplement 1B***) we observed drastically impaired phosphorylation of T$_{184}$-Rasgrp1, and reduced ERK phosphorylation in *Rasgrp1^Anaef* thymocytes (***Figure 5A***). By contrast, *Rasgrp1^+/−* thymocytes heterozygous for the null allele displayed readily detectable Rasgrp1- and ERK- phosphorylation (***Figure 5B***), demonstrating that the signaling defects in the *Rasgrp1^Anaef* thymocytes are much greater than when the amount of

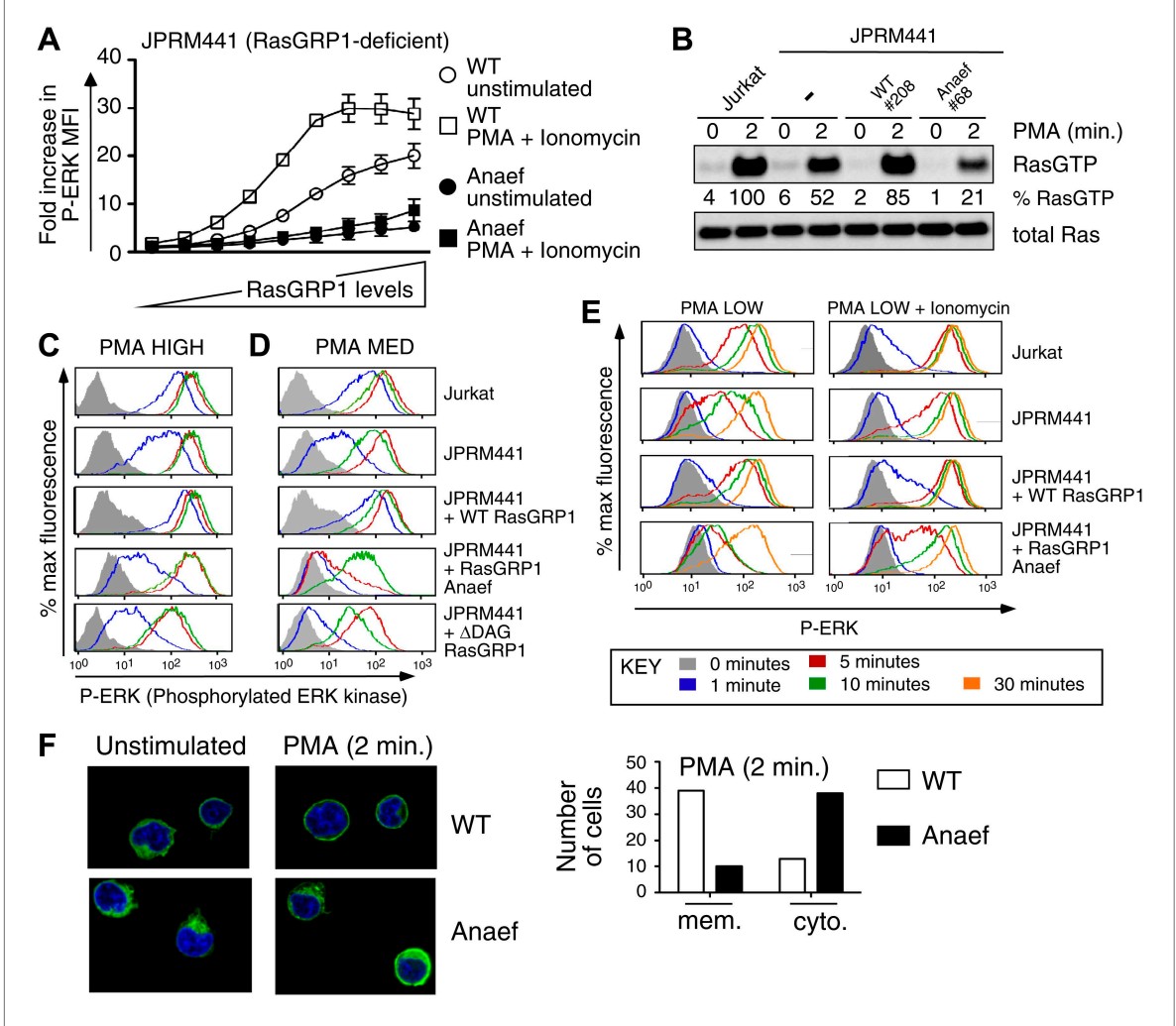

**Figure 4**. Rasgrp1$^{Anaef}$ diminishes in vitro signaling to Ras and ERK. (**A**) RasGRP-1 deficient Jurkat cells (JPRM441) were transiently transfected with EGFP-tagged wildtype- or Anaef-RasGRP1 and RasGRP1-dose/P-ERK responses were determined by FACS. (**B**) Jurkat, JPRM441, and JPRM441 cells stably reconstituted with Anaef- or WT Rasgrp1 were left unstimulated or stimulated with 25 ng/ml PMA and RasGTP levels were determined through a RasGTP pull down assay. (**C–E**) The indicated cell lines were stimulated for 0, 1, 5, 10, or 30 min (see key) with a high (25 ng/ml, **C**), medium (5 ng/ml, **D**), or low dose (2 ng/ml, **E**) of PMA, or with 2 ng/ml PMA with ionomycin (1 µM) and stained for intracellular phosphorylated-ERK-1 and -2 (P-ERK), and analyzed by flow cytometry. Results for JPRM441-Rasgrp1Anaef clones #68 are shown, similar results were obtained with clone #69. (**F**) EGFP-tagged constructs were transiently transfected into JPRM441 and RasGRP1 localization was determined for unstimulated and PMA-stimulated cells as described in the text. 50 images for the 2-min PMA stimulation were scored in a blinded manner and plotted with RasGRP1-Anaef in the black columns.
The following figure supplements are available for figure 4:

**Figure supplement 1**. Biochemical aspects of the *Rasgrp1$^{Anaef}$* allele.

Rasgrp1 is simply halved. Thymocyte subset-specific P-ERK analyses revealed reduced anti-CD3- and PMA-induced responses in DP, CD4SP, and CD8SP *Rasgrp1$^{Anaef}$* thymocytes (**Figure 5C**, **Figure 5—figure supplement 1C**), echoing the cell line conclusion that the *Anaef* mutation results in a partial loss of function with respect to induced Ras-ERK signaling. When pressure was placed on TCR-Ras-ERK signaling for positive selection in vivo, by introducing any one of three different rearranged TCR transgenes that are prematurely expressed at higher than normal levels on DP thymocytes, a small decrease in positive selection was revealed in *Rasgrp1$^{Anaef}$* TCR-transgenic thymocytes compared to their wildtype controls (**Figure 5—figure supplement 2**). Collectively, these results lead to the surprising conclusion that the low affinity pMHC stimulation that drives physiological positive selection in vivo is remarkably robust to decreased Rasgrp1 activation of Ras-ERK.

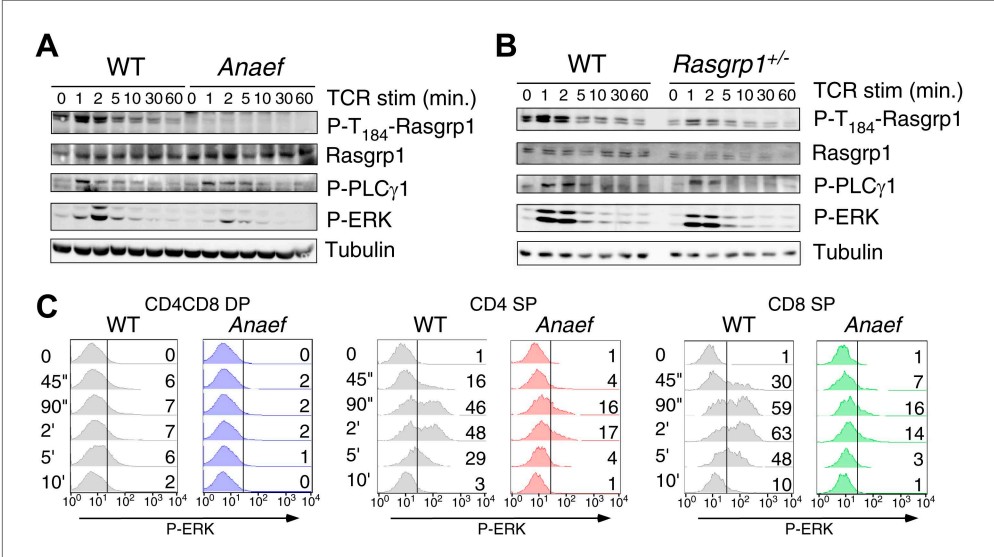

**Figure 5**. *Rasgrp1^Anaef* signaling characteristics of in vitro stimulated thymocytes. (**A** and **B**) Phosphorylation of PLCγ1, Rasgrp1, and ERK in total thymocytes from *Rasgrp1^Anaef* and age-matched WT mice or *Rasgrp1^{+/−}* and age-matched WT mice stimulated with anti-CD3 antibodies. (**C**) P-ERK induction by in vitro stimulation followed by intracellular flow cytometric staining and electronic gating on thymocyte subsets. Thymocytes from *Rasgrp1^Anaef* and age-matched WT mice were stimulated with anti-CD3 antibody (5 μg/ml) and crosslinked using secondary antibody (Goat anti-hamster, 20 μg/ml) for the indicated time points, fixed, and stained for intracellular p-ERK. Histograms show P-ERK staining on electronically gated DP, CD4SP or CD8SP thymocytes. These conditions yielded low-levels of ERK phosphorylation that is typically observed for wildtype DP thymocytes. Numbers indicate the percentages of cells above the arbitrarily set reference point in the histograms so that it can be appreciated how many cells within each population cross this P-ERK threshold. Data is representative of three independent experiments. For PMA-induced responses see *Figure 5—figure supplement 1D*.

The following figure supplements are available for figure 5:

**Figure supplement 1**. Signaling characteristics of *Rasgrp1^Anaef* thymocytes.

**Figure supplement 2**. Thymic selection in *Rasgrp1^Anaef* mice carrying transgenic TCRs.

## Intrinsically dysregulated formation of Helios+ PD-1+ CD4+ T cells in *Rasgrp1^Anaef* mice

Given the evidence above for normal thymic formation of T cells in *Rasgrp1^Anaef* animals with normal TCR genes, we sought to define the peripheral CD4+ cell dysregulation that results in an expanded population of CD44^hi CD4+ T cells. Total splenocyte numbers and CD4 subsets were within the normal range in young *Rasgrp1^Anaef* animals, but between 50 and 150 days of age the frequencies of activated or memory CD44^hi Foxp3− CD4+ cells increased, as did Foxp3+ CD4 cells, while the frequency of CD44^low Foxp3− naïve CD4+ cells decreased (*Figure 6A–C*).

Further resolution of CD4+ subsets based on intracellular cytokine staining revealed that interferon-γ producing cells were increased in frequency by a similar magnitude as CD44^hi cells as a whole (*Figure 6D* and data not shown). Thus, there was no evidence that the *Rasgrp1^Anaef* mutation skewed T-helper cells towards a Th1 phenotype, but simply increased the number of activated or memory/effector CD4+ cells. Staining for PD-1 and CXCR5, whose high expression on CD4+ cells identifies T follicular helper (T_{FH}) cells (*Ramiscal and Vinuesa, 2013*) revealed a dramatic expansion of these cells in *Rasgrp1^Anaef* mice (*Figure 6E*). However most of the increase in CD44^hi Foxp3− CD4+ cells in *Rasgrp1^Anaef* mice was due to a 600% increase in cells that expressed intermediate levels of PD-1 and CXCR5, and hence are unlikely to be T_{FH} cells, but were distinguished by high expression of the Helios transcription factor (*Figure 6E*). Helios is highly expressed in Foxp3+ T-reg cells (*Thornton et al., 2010*), but in wildtype and *Rasgrp1^Anaef* mice Helios is also upregulated in a subset of Foxp3− CD4+ cells, nearly all of which are CD44^hi (*Figure 6F*). *Rasgrp1^Anaef* greatly increased this Helios+ CD44^hi Foxp3− CD4+ population,

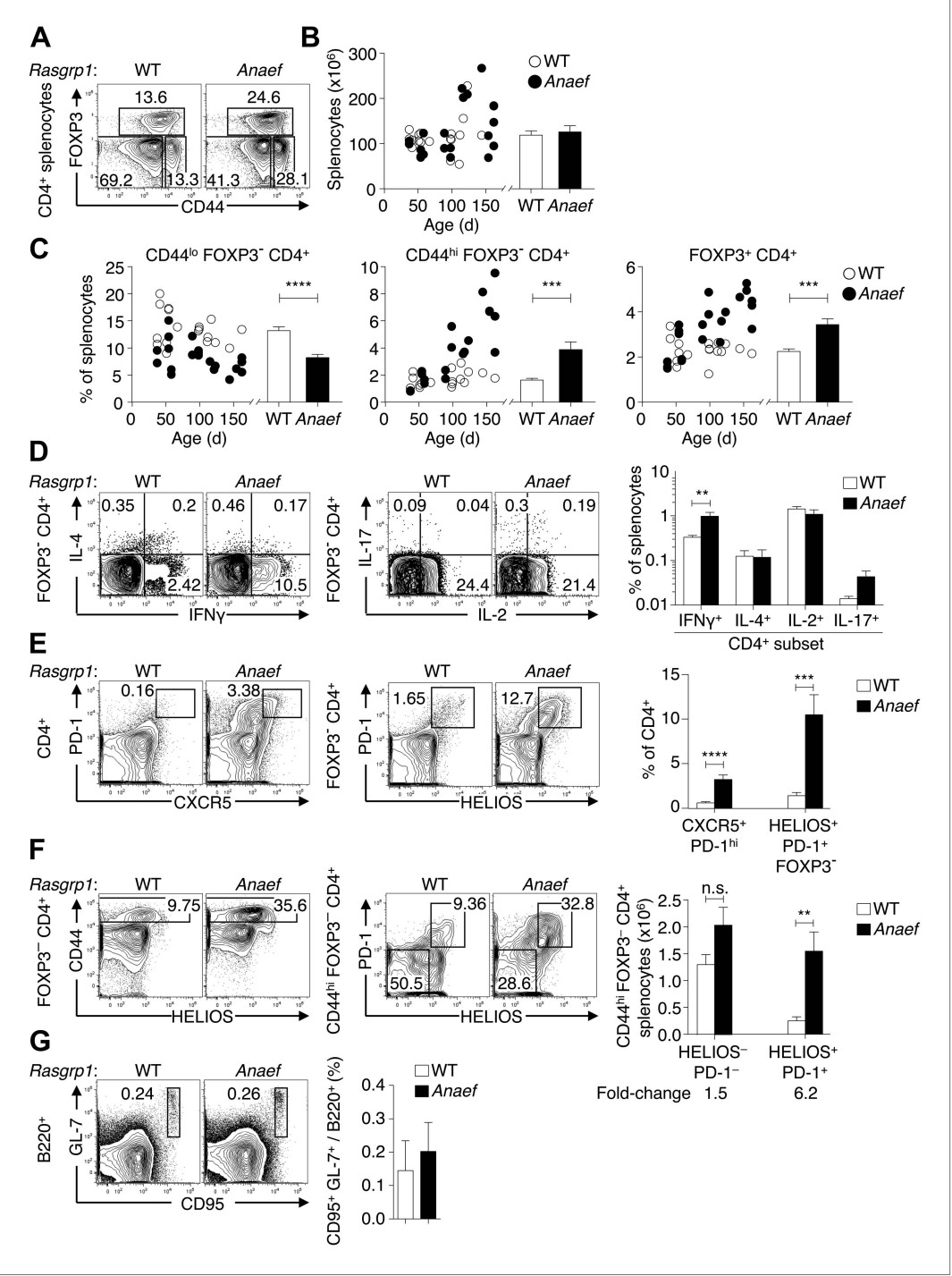

**Figure 6**. *Rasgrp1^Anaef* results in dysregulation of peripheral CD4 T cells. (**A**) Representative plots of wildtype or *Rasgrp1^Anaef* CD4+ splenocytes subsetted into naïve (CD44^lo FOXP3−), activated/memory (CD44^hi FOXP3−), and T-reg (FOXP3+) populations. (**B** and **C**) Splenic cellularity and frequencies of the CD4+ subsets gated in (**A**) as a function of age in wildtype vs *Rasgrp1^Anaef* mice; each dot represents one mouse (WT in white; *Rasgrp1^Anaef* in black). Inset column graphs show the group mean ± SEM. Statistics obtained by unpaired Student's *t* test. ***p<0.001, ****p<0.0001. (**D**) Representative intracellular labeling of IFNγ, IL-4, IL-2 or IL-17 on electronically gated Foxp3− CD4+ wildtype or *Rasgrp1^Anaef* splenocytes that had been stimulated with PMA and ionomycin for 4 hr. Column graphs show mean ± SEM frequencies amongst all splenocytes. Statistical analysis of % IFNγ+ cells used an unpaired Student's *t* test (n = 7 WT, 6 *Anaef*) **p<0.01. (**E**) Phenotype of CD4+ splenocytes showing a CXCR5+ PD-1^hi gate for

*Figure 6. Continued on next page*

*Figure 6. Continued*

the T$_{FH}$ population (left) and a gate for the HELIOS$^+$ PD-1$^+$ population amongst Foxp3$^-$ CD4$^+$ splenocytes (right). Column graph shows mean ± SEM frequencies of these populations amongst CD4$^+$ splenocytes in wildtype or *Rasgrp1$^{Anaef}$* mice. Statistical analyses used unpaired Student's *t* tests (n = 19 WT, 18 *Anaef*). ***p<0.001, ****p<0.0001. (**F**) Helios vs CD44 phenotype of Foxp3$^-$ CD4$^+$ splenocytes (left plots) showing the gate for the CD44$^{hi}$ population, which was analyzed for expression of Helios and PD-1 (right plots). Column graph shows the mean ± SEM number of splenocytes within the CD44$^{hi}$ Foxp3$^-$ CD4$^+$ subpopulations gated in the right plots (n = 19 WT, 20 *Anaef* mice aged >70 days and compiled from 11 separate experiments). Unpaired Student's *t* test **p<0.01. (**G**) Phenotype of B220$^+$ splenocytes showing the CD95(Fas)$^{hi}$ GL-7$^{hi}$ gate used to define germinal center B cells (left), the mean ± SEM frequency of which is shown in the column graph (right) (n = 7 WT, 5 *Anaef*).

which was also distinguished by high PD-1 expression in *Rasgrp1$^{Anaef}$* (**Figure 6F**). *Rasgrp1$^{Anaef}$* mice had normal frequencies of CD95 (Fas)$^+$ GL-7$^+$ germinal center B cells in the spleen (**Figure 6G**), consistent with the conclusion that the accumulating Helios$^+$ PD-1$^+$ CXCR5$^{int}$ Foxp3$^-$ CD44$^{hi}$ CD4$^+$ cells were a distinct type of activated CD4$^+$ cell but not fully differentiated T$_{FH}$ cells.

To test if the dysregulated accumulation of Helios$^+$ PD-1$^+$ CXCR5$^{int}$ CD44$^{hi}$ CD4$^+$ T cells in *Anaef* mice required B cells, *Rasgrp1$^{Anaef}$* animals were intercrossed with mice bearing a null mutation in the BCR subunit, CD79a (**Yabas et al., 2011**). In *Rasgrp1$^{Anaef}$ Cd79a$^{null}$* animals lacking B cells, accumulation of PD-1$^+$ Helios$^+$ CD4$^+$ T cells was profoundly suppressed (**Figure 7A,B**). Indeed, the *Rasgrp1$^{Anaef}$*–driven distortion in relative frequencies of naïve, effector/memory and regulatory subsets of CD4 splenocytes was rectified by the absence of B cells in *Rasgrp1$^{Anaef}$ Cd79a$^{null}$* mice (**Figure 7C,D**). By contrast, the elevated CD44 expression on naïve CD4$^+$ cells was still present (**Figure 7E**) indicating this is a constitutive effect of the *Rasgrp1$^{Anaef}$* mutation.

The requirement for B cells could indicate they are needed as specialized antigen presenting cells, as is the case for T$_{FH}$ cells (**Ramiscal and Vinuesa, 2013**), or that the *Anaef* mutation also acts in B cells since B cells also express Rasgrp1 (**Stone, 2011**). To resolve these alternatives, we used bone marrow from *Rasgrp1$^{Anaef}$ Cd79a$^{null}$* animals mixed with wildtype *Rasgrp1$^{+/+}$* marrow to generate chimeric mice where the *Rasgrp1$^{Anaef}$* mutation was excluded from B cells but present in most of the T cells (experimental group B in **Figure 7F–I**), and compared these with control chimeras where all hematopoietic cells were *Rasgrp1$^{Anaef}$* or *Rasgrp1$^{WT}$* (groups A, C and D in **Figure 7**). Despite having the *Anaef* mutation in the T but not B cells of group B mice, a high proportion developed antinuclear autoantibodies comparable to the control group D where both B and T cells carried the *Anaef* mutation (**Figure 7G**). Moreover, a high frequency and number of CD45.2$^+$ *Rasgrp1$^{Anaef}$* CD4$^+$ T cells acquired a Helios$^+$ PD-1$^+$ phenotype in Group B animals, unlike the co-resident CD45.1$^+$ wildtype T cells (**Figure 7I**). The accumulation of these activated CD4 T cells thus reflects a cell-autonomous effect of the *Anaef* mutation within the CD4 T cells and does not depend upon the *Anaef* allele being present in B cells.

## CD44 expression is a sensitive reporter of mTOR activity in naïve T cells

Increased basal expression of the cell adhesion receptor, CD44, was a unique trait exhibited by naïve, CD62L-positive *Rasgrp1$^{Anaef}$* T cells (**Figure 1B**). CD44 expression normally increases during differentiation of DP thymocytes into SP T cells, attains higher levels on naïve CD4 T cells than on naïve CD8 cells, and increases further on activated/memory T cells, but little was known about what determines the level of CD44 expressed. In cancer cells CD44 has been described as an mTOR target (**Hsieh et al., 2012**). In our ongoing peripheral blood screen of ENU mutagenized mouse pedigrees, we identified a strain, *chino*, with decreased CD44 expression on peripheral CD4$^+$ CD62L$^{hi}$ cells but relatively normal T cell numbers and subsets (**Figure 8A**). This unusual phenotype mapped to a single nucleotide change (T to G) in exon 5 of the mechanistic target of rapamycin (*Mtor*) gene, introducing serine in place of isoleucine at position 205 in the fifth predicted HEAT domain of the protein (**Knutson, 2010**) (**Figure 8B**). This mTOR isoleucine residue is entirely conserved from mammals to yeast (**Figure 8—figure supplement 1**). The mTOR HEAT-repeat domain forms a large superhelical structure that binds RAPTOR to recruit substrates such as S6 kinase for phosphorylation by the mTOR kinase domain (**Kim et al., 2002**; **Adami et al., 2007**).

As the absence of *Mtor* is embryonic lethal in mice (**Gangloff et al., 2004**; **Murakami et al., 2004**), the fact that *Mtor$^{chino/chino}$* (hereafter referred to as *Mtor$^{chino}$*) mice are viable but slightly smaller than wildtype (**Figure 8C**) indicates that the *Mtor$^{chino}$* allele retains substantial function. Consistent with a

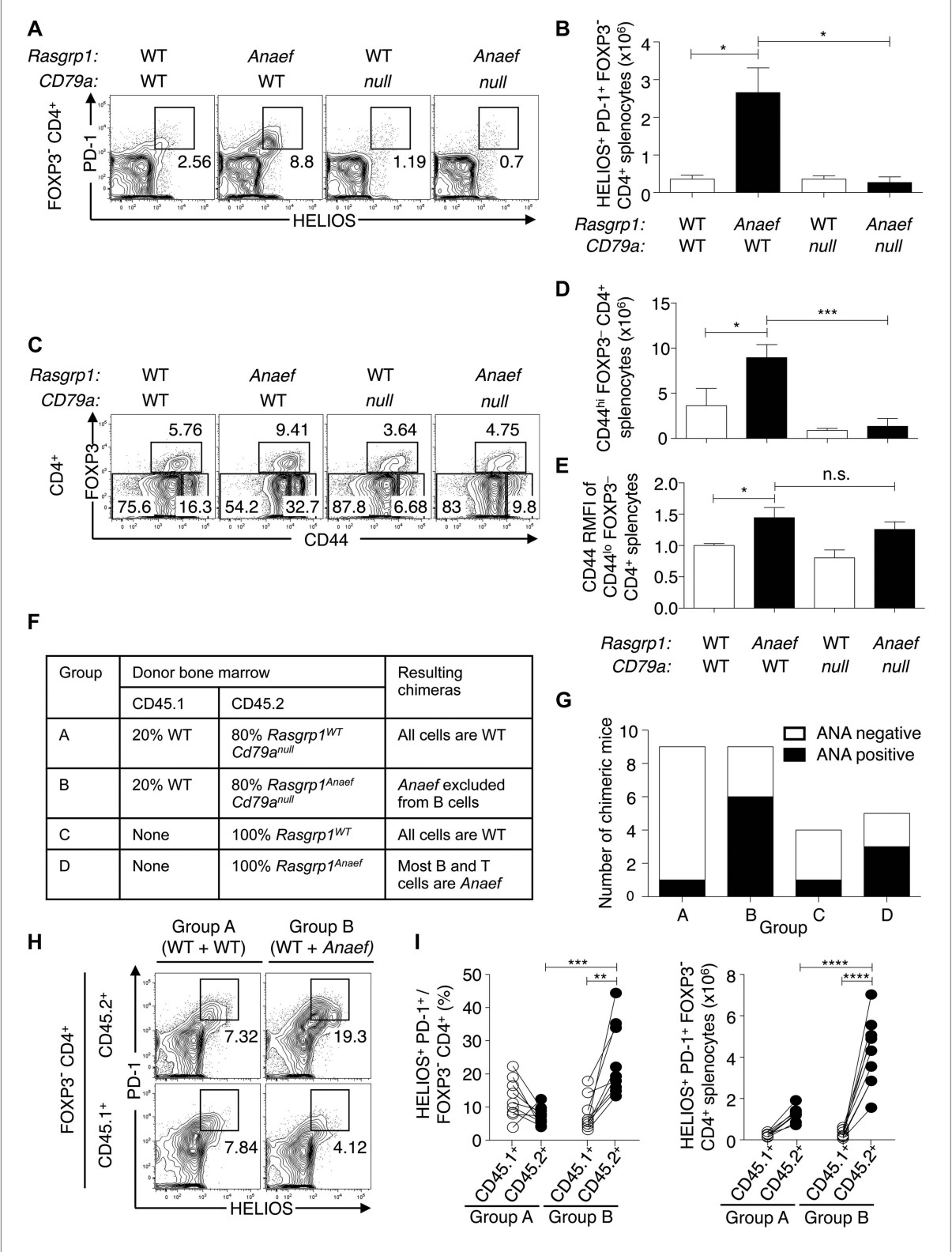

**Figure 7**. Role of B and T cells in *Rasgrp1^Anaef*-induced Helios+ PD-1+ CD4+ T cell and autoantibody formation. (**A–E**) B6.*Cd79a^−/−* mice lacking B cells were intercrossed with B6.*Rasgrp1^Anaef* mice to produce mice with the genotypes shown above the plots. Plots (left) display phenotype of (**A** and **B**) Foxp3− CD4+ splenocytes and the gate used to define the PD-1+ HELIOS+ subpopulation and absolute numbers of these cells are shown on the column graphs (mean ± SEM). Statistical comparisons used unpaired Student's *t* tests (n = 4 WT, 5 *Anaef*, 2 *CD79a-null*, and 4 *Anaef.CD79a-null*) *p<0.05. (**C**) CD4+ splenocytes

*Figure 7. Continued on next page*

*Figure 7. Continued*

and the gates used to define FOXP3⁺ regulatory, CD44^hi FOXP3⁻ effector/memory, and CD44^lo FOXP3⁻ naïve subsets. Absolute numbers of effector/memory splenocytes (**D**) and Relative CD44 expression on naive splenocytes (**E**) are shown in the column graphs (mean ± SEM). Statistical comparisons used unpaired Student's *t* tests (n = 4 WT, 5 *Anaef*, 2 *CD79a-null*, and 4 *Anaef.CD79a-null*) *p<0.05, ***p<0.001. (**F**) Experimental design to delineate role of *Rasgrp1^Anaef* in T and B cells. Irradiated mice received either 100% *Rasgrp1^WT* marrow, 100% *Rasgrp1^Anaef* marrow, or a 1:4 mixture of CD45.1⁺ wild-type marrow mixed with CD45.2⁺ marrow from either *Cd79a^null/null Rasgrp1^WT* or *Cd79a^null/null Rasgrp1^Anaef* siblings. (**G**) Incidence of homogeneous nuclear ANA in blood plasma in chimeric mice collected 18 weeks after irradiation, measured by immunofluorescence on HEp-2 cells and scored in a blinded manner. (**H**) Representative plots (left), and (**I**) quantification of frequency (middle) and number (right) of PD-1⁺ Helios⁺ cells among Foxp3⁻ CD4⁺ splenocytes from mixed chimeras in groups A and B as described in (**F**) 45 weeks after irradiation. Lines in (**I**) connect measurements from individual mice. Statistical analysis used paired Student's *t* tests within groups and unpaired *t* tests between groups. **p<0.01, ***p<0.001, ****p<0.0001.

subtle decrease in mTOR activity, TCR-induced phosphorylation of ribosomal protein S6 (P-S6) was modestly decreased but not abolished in *Mtor^chino* CD4⁺ splenocytes (*Figure 8D*). The numbers of DP, CD4SP and CD8SP thymocytes were normal in *Mtor^chino* mice (*Figure 8E*). Whereas expression of CD69, CD5 and TCRβ on DP and SP thymocytes was normal, CD44 expression was decreased on *Mtor^chino* CD4SP and CD8SP thymocytes (*Figure 8F*). CD44 expression on peripheral blood CD4⁺ T cells decreased in an *Mtor^chino* allele dose-dependent manner (*Figure 8G*), demonstrating that CD44 expression is a highly sensitive reporter of small changes in basal mTOR activity in CD4⁺ T cells. This conclusion is reinforced by supplementary data from two studies: mice with a *neo*-insertion in an *Mtor* intron that decreases *Mtor* mRNA approximately 70% show a similar selective lowering of CD44 on CD4⁺ T cells (*Zhang et al., 2011*) and CD44 levels are also reduced on naïve, CD62L-positive T cells that are deficient for the mTOR activator Rheb or deficient for the mTOR-interacting protein Rictor (*Delgoffe et al., 2011*). Furthermore, we found that pharmacological inhibition of mTOR with rapamycin in mice treated with low doses that avoid toxicity (*Coenen et al., 2007*; *Araki et al., 2009*), resulted in a dose-dependent decrease in CD44 expression on wildtype and *Rasgrp1^Anaef* thymocytes (*Figure 8H*).

### *Rasgrp1^Anaef* exaggerates basal mTOR-S6 signaling and CD44 expression in T cells

Our finding that tonic CD44 expression on naïve T cells sensitively reports small changes in mTOR activity prompted further analysis of this pathway in *Rasgrp1^Anaef* T cells. CD44 was elevated on *Rasgrp1^Anaef* DP and SP thymocytes, in diametric contrast to decreased CD44 on these cells in *Rasgrp1^−/−* knockout mice (*Figure 9A,B*) (*Priatel et al., 2007*). A putative defect in regulatory T cells cannot explain the increased CD44 expression on naïve *Rasgrp1^Anaef* CD4⁺ cells, because in mixed chimeras bearing many wild-type Foxp3⁺ CD4 cells, CD44 expression was still increased on *Rasgrp1^Anaef* but not on co-resident wild-type CD62L⁺ FOXP3⁻ CD4⁺ splenocytes (*Figure 9C*). The cell autonomous increase in CD44 expression was detectable on *Rasgrp1^Anaef* SP and DP thymocytes (*Figure 9—figure supplement 1A,B*) and even on CD4⁺ cells expressing a transgenic TCR (*Figure 9D*). Elevated CD44 expression on *Rasgrp1^Anaef* cells was selective: there were no distinguishable differences between wildtype and *Rasgrp1^Anaef* cells in CD69 and TCRβ, which are markers of cumulative Ras signaling (*D'Ambrosio et al., 1994*; *Genot and Cantrell, 2000*; *Starr et al., 2003*), nor in CD5 expression, a sensitive reporter of TCR affinity and constitutive or tonic TCR signaling (*Azzam et al., 1998*; *Mandl et al., 2013*) (*Figure 9E*, *Figure 9—figure supplement 1A,B*), and rapamycin treatment in vivo selectively reduced the increased CD44 expression on *Rasgrp1^Anaef* cells (*Figure 8H*) but did not impact TCR or CD69 expression (data not shown).

Consistent with the CD44 data, P-S6 levels in unstimulated TCRβ^low (pre-selection) DP thymocytes were modestly increased in *Rasgrp1^Anaef* mice whereas they were decreased in cells from *Rasgrp1^−/−* mice (*Figure 10A*). CD44 levels gradually rise as T cells mature from DP to SP and peripheral naïve T cells, suggesting that CD44 expression may reflect basal or tonic signaling. Such signals can be visualized by constitutive tyrosine-phosphorylation of the TCRζ chains and other proteins (*van Oers et al., 1993*), basal levels of ZAP70 recruitment to phosphorylated zeta chains (*van Oers et al., 1994*; *Stefanova et al., 2002*), and tonic phosphorylation of ERK (*Roose et al., 2003*; *Markegard et al., 2011*). Furthermore, these signals dissipate when T cells are rested in vitro in non-stimulatory medium (*van Oers et al., 1993*; *Stefanova et al., 2002*). We investigated tonic mTOR–S6 signals and found substantial basal P-S6 levels in freshly isolated lymph node cells that decreased when cells were serum-starved in vitro

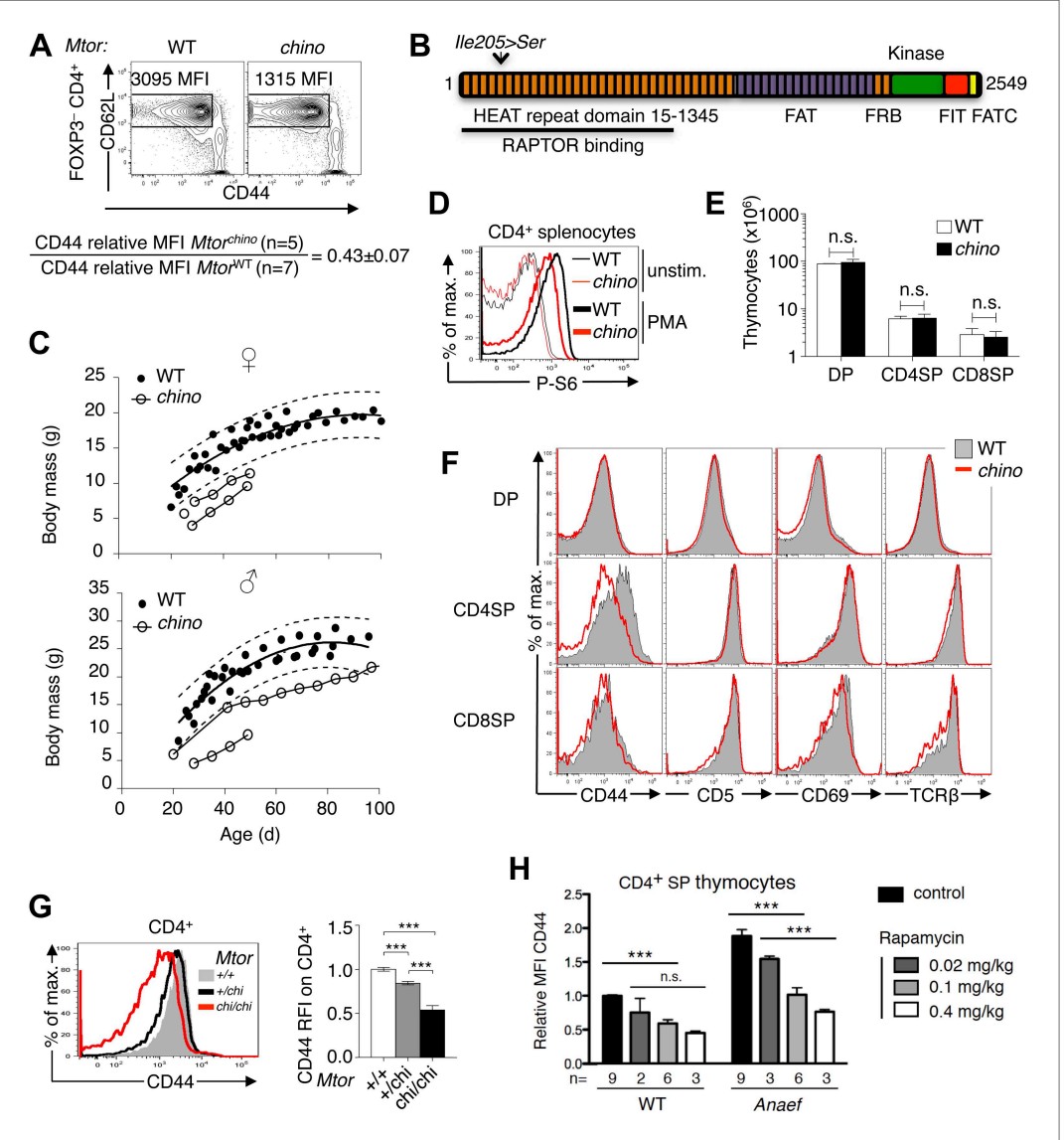

**Figure 8**. CD44 is a sensitive reporter of mTOR activity in immature and naïve CD4+ T cells. (**A**) CD44lo T cell phenotype upon which *Mtor*chino mice were identified. Phenotype of Foxp3– CD4+ splenocytes from *Mtor*+/+ (WT) and *Mtor*chino/chino (*chino*) mice, showing the CD62Lhi subset on which CD44 expression was quantified and normalized to the mean of WT animals (mean ± SEM from three experiments shown below). (**B**) Schematic of mTOR protein showing functional domains and the *chino* I205S mutation in the fifth HEAT repeat. (**C**) Reduced body size in *chino*. Body mass vs age of WT (black dots; 10 female [top], 11 male [bottom]) or *chino* (unfilled dots; three female, two male) mice. Curves were fitted to the WT datasets using second order polynomial equations; dotted lines show 95% prediction bands (the area expected to enclose 95% of future WT data points; GraphPad Prism version 5.0d for MAC OS X). Straight lines connect multiple measurements of individual *chino* mice. (**D**) Splenocytes from WT or *chino* mice were left unstimulated or were stimulated with PMA (100 ng/ml) for 10 min, fixed, and stained for intracellular phosphorylated-S6 (P-S6). Histogram overlay shows P-S6 staining on CD4+ cells representative of two separate experiments. (**E**) Number of DP, CD4SP and CD8SP cells per thymus of WT (n = 7) or *chino* (n = 7) mice compiled from four experiments. (**F**) Selective reduction in CD44 expression on *chino* thymocytes. Histograms show CD44, CD5, CD69 or TCRb expression on gated DP, CD4SP or CD8SP thymocytes from WT (solid gray) vs *chino* (red overlay) mice, representative of four separate experiments (n = 7 mice per group in total). (**G**) Histogram (left) and column graph (right, mean ± SEM) shows CD44 expression on CD4+ CD3+ B220– peripheral blood lymphocytes from littermate mice of the indicated *Mtor* genotypes, compiled from five separate experiments using a total of 20 *Mtor*+/+, 49 *Mtor*+/chi and 7 *Mtor*chi/chi mice. CD44 relative fluorescence intensity (RFI) was calculated by dividing by the mean for the *Mtor*+/+ group analyzed in the same experiment. Unpaired Student's *t* tests: ***p<0.0001. (**H**) CD44
*Figure 8. Continued on next page*

*Figure 8. Continued*

RFI on unstimulated CD4SP thymocytes of WT or *Rasgrp1^Anaef* mice after 7 days of treatment with the indicated rapamycin doses. Untreated control was set at 1. ***p<0.005, n.s. = non significant.

The following figure supplements are available for figure 8:

**Figure supplement 1**. Alignment of HEAT5 domains in mTOR.

(*Figure 10B*). A recent study reported that naive CD4$^+$ T cells display a range of CD5 expression in which the CD5$^{high}$ cells receive most tonic signal input and are most immune reactive (*Mandl et al., 2013*). We first sorted CD44$^{low}$ naïve CD4$^+$ T cells into the most bright and most dim expression for CD5 and determined that CD5$^{high}$ naïve CD4$^+$ T cells have significantly more P-S6 than their CD5$^{low}$ counterparts (*Figure 10C*). Next, dividing CD5$^{low}$ and CD5$^{high}$ naïve CD4$^+$ T cells in equal 50–50% splits revealed that basal P-S6 was increased in *Rasgrp1^Anaef*, particularly in the CD5$^{low}$ subset compared to wildtype cells (*Figure 10D*). The exact origin of tonic signals in T cells and its function being either immune stimulatory or immune suppressive is still an area of debate (*Polic et al., 2001*; *Smith et al., 2001*; *Bhandoola et al., 2002*; *Stefanova et al., 2002*; *Hogquist et al., 2003*), but at least part of the tone appears to be generated by low affinity TCR binding to self pMHC (*Stefanova et al., 2002*). To examine if self-peptide/MHCII recognition plays a role in the increased CD44 expression on *Rasgrp1^Anaef* naïve CD4$^+$ T cells, we adoptively transferred a mixture of wild-type and *Rasgrp1^Anaef* splenocytes into wild-type or MHCII(*H2-Aa*)–deficient recipient mice (*Figure 10—figure supplement 1*). Maintenance of CD5 expression on T cells requires contact with self pMHC (*Smith et al., 2001*; *Mandl et al., 2012*) and, as expected, CD5 expression on wildtype CD4$^+$ CD62L$^+$ Foxp3$^-$ T cells decreased in MHCII-deficient hosts (*Figure 10E*), consistent with the hypothesis that CD5 is a sensitive reporter of TCR signal strength. By contrast, CD44 levels were similar irrespective of MHCII expression in the adoptive hosts, and CD44 levels remained higher on *Rasgrp1^Anaef* than co-transferred wild-type cells in both contexts (*Figure 10E*). These data reveal constitutively increased expression of two reporters of mTOR activity in *Rasgrp1^Anaef* naïve CD4$^+$ T cells: the well-established reporter P-S6 and the reporter clarified here, CD44. The fact that the *Rasgrp1^Anaef*–driven increase in CD44 is retained in the absence of MHCII suggests that this tonic signal is at least partially independent of triggering of TCRs by self-pMHC.

### CD44$^+$ Helios$^+$ PD-1$^+$ CD4$^+$ T cell accumulation and autoantibodies in *Rasgrp1^Anaef* mice are corrected by hypomorphic mTOR mutation

To test the role of elevated mTOR signaling in the *Rasgrp1^Anaef*–induced overexpression of CD44 in naïve T cells and in the accumulation of activated CD44$^{hi}$ PD-1$^+$ CD4$^+$ cells and autoantibodies, *Rasgrp1^Anaef* mice were intercrossed with the subtle loss-of-function *Mtor^chino* strain. Using a CD62L$^+$ Foxp3$^-$ gate to resolve naïve CD4$^+$ splenocytes, we found that the *Mtor^chino* mutation abolished the *Rasgrp1^Anaef*–driven increase in CD44 expression in these naïve T cells (*Figure 11A*). The *Mtor^chino* mutation alone resulted in a decrease in numbers of splenocytes, including CD4$^+$ cells, as was observed in mice with reduced *Mtor* mRNA (*Zhang et al., 2011*), but the relative proportions of the CD4$^+$ subsets examined were normal (*Figure 11B*). Whereas *Rasgrp1^Anaef* mice with normal mTOR accumulated a high frequency of CD44$^{hi}$ Foxp3$^-$ CD4$^+$ splenocytes, including the prominent PD-1$^+$ Helios$^+$ subset, this was corrected down to normal numbers in *Rasgrp1^Anaef* *Mtor^chino* double mutants (*Figure 11B,C*). Moreover, in bone marrow chimeras bearing *Rasgrp1^Anaef* *Mtor^chino* double-mutant hematopoietic cells, the frequency of animals with antinuclear autoantibodies was corrected to the low frequency observed in control chimeras with wildtype *Rasgrp1* and *Mtor* (*Figure 11D*). Collectively, these results establish that accumulation of CD44$^{hi}$ Helios$^+$ PD-1$^+$ CD4$^+$ cells and autoantibodies induced by *Rasgrp1^Anaef* is sensitive to small differences in mTOR signaling.

## Discussion

The findings here reveal a new role for Rasgrp1 in the cell-intrinsic regulation of peripheral CD4$^+$ T cells. By analyzing a missense mutation in the Rasgrp1 EF-hand, the results dissociate this new function of Rasgrp1 from its well-known role in thymic positive selection. While the *Anaef* EF-hand mutation did decrease Rasgrp1 activation of Ras and ERK when thymocytes were stimulated acutely by antibodies to CD3 or with PMA in vitro, the activity of this pathway during physiological positive selection in vivo

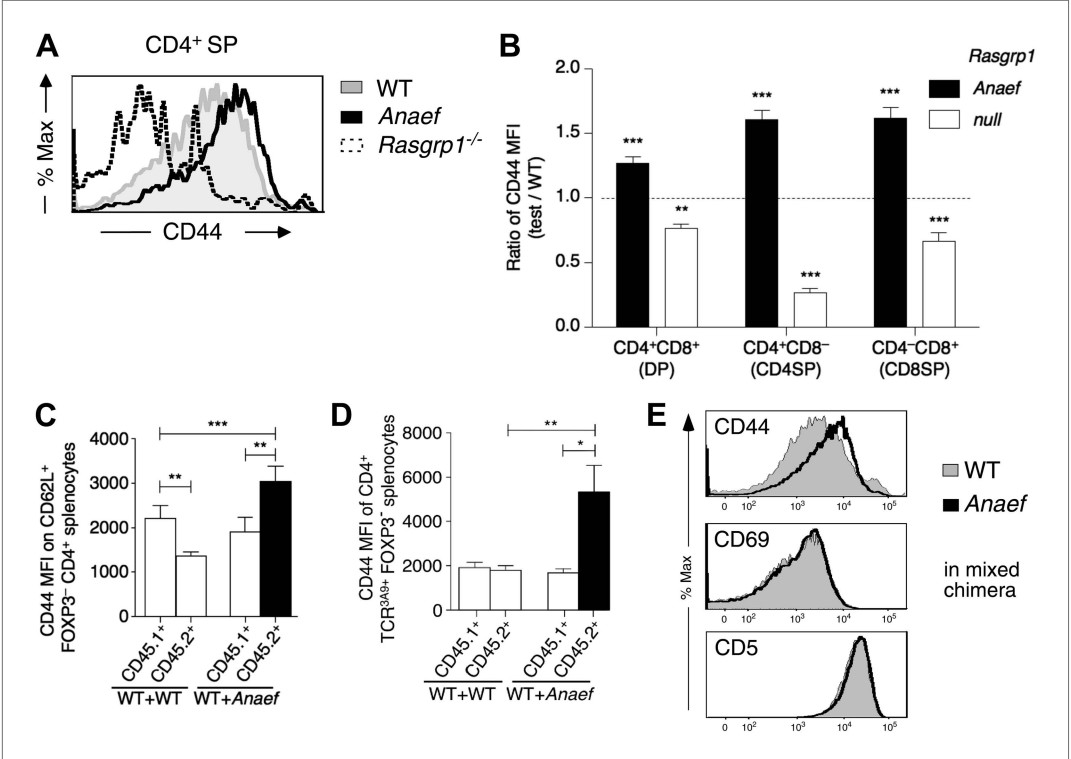

**Figure 9**. Selective and cell-autonomous increase in CD44 expression in *Rasgrp1^Anaef^* CD4+ T cells. (**A**) Representative CD44 expression on unstimulated CD4SP thymocytes from wildtype, *Rasgrp1^Anaef^* and *Rasgrp1^−/−^* mice. (**B**) Mean relative CD44 from 22 *Rasgrp1^Anaef^*, 13 wildtype, and 3 *Rasgrp1^−/−^* mice compiled from seven experiments. In each experiment, the mean CD44 expression for the wildtype group was taken to be one, and values for individual mice were normalized to this. Unpaired Student's *t* tests were used to compare *Rasgrp1^Anaef^* and *Rasgrp1^−/−^* with the wildtype group. **p<0.005, ***p<0.0005. (**C**) CD44 MFI on CD62L^+^Foxp3^−^CD4^+^ splenocytes from mixed bone marrow chimeras described in **Figure 7F**. Statistical comparisons used paired *t* tests within, and unpaired *t* tests between, groups of chimeras (n = 9 for both groups); ***p<0.001, **p<0.01. (**D**) CD44 MFI on TCR3A9+ (clonotype positive) CD4^+^Foxp3^−^ splenocytes from B10.BR mixed bone marrow chimeras containing CD45.1^+^ wildtype TCR3A9 plus either CD45.2^+^ wildtype (n = 8) or CD45.2^+^ *Rasgrp1^Anaef^* (n = 7) 3A9 TCR-transgenic hematopoietic cells. Statistical comparisons in (**C**) and (**D**) used paired *t* tests within groups of chimeras and unpaired *t* tests between groups of chimeras. p value symbols: ***p<0.001, **p<0.01, *p<0.05. (**E**) *Rasgrp1^Anaef^* increases CD44 expression levels but does not affect CD69 or CD5 expression. CD4SP thymocytes are analyzed from irradiated mice reconstituted with non-transgenic CD45.1^+^ wildtype mixed with CD45.2+ Rasgrp1^Anaef^ bone marrow.

The following figure supplements are available for figure 9:

**Figure supplement 1**. CD44 and P-S6 expression in thymocytes.

remained sufficient for normal numbers of single positive thymocytes and peripheral T cells to form even under competitive reconstitution conditions. T cell lymphopenia and sparse T cell repertoires secondary to defective positive selection potentially explain the autoantibodies observed in mice where Rasgrp1 is entirely absent or C-terminally deleted (*Dower et al., 2000*; *Coughlin et al., 2005*; *Priatel et al., 2007*; *Fuller et al., 2012*), and in animals with missense mutations in ZAP-70 or LAT (*Aguado et al., 2002*; *Sommers et al., 2002*; *Sakaguchi et al., 2003*; *Siggs et al., 2007*). By contrast, the normal thymic development coupled with experiments in mixed bone marrow chimeras where wild-type and mutant T cells co-exist rules out this possibility for the *Rasgrp1^Anaef^* mutation, and shows that peripheral CD4 cells are intrinsically dysregulated. By a combination of biochemical and genetic studies, we identify overactive mTOR signaling within naïve CD4 T cells as a key component for *Rasgrp1^Anaef^* to drive two abnormalities: (1) a constitutive increase in CD44 expression in naïve CD4 T cells and (2) a gradual accumulation of peripheral Helios^+^ CD44^hi^ CD4 cells and autoantibodies.

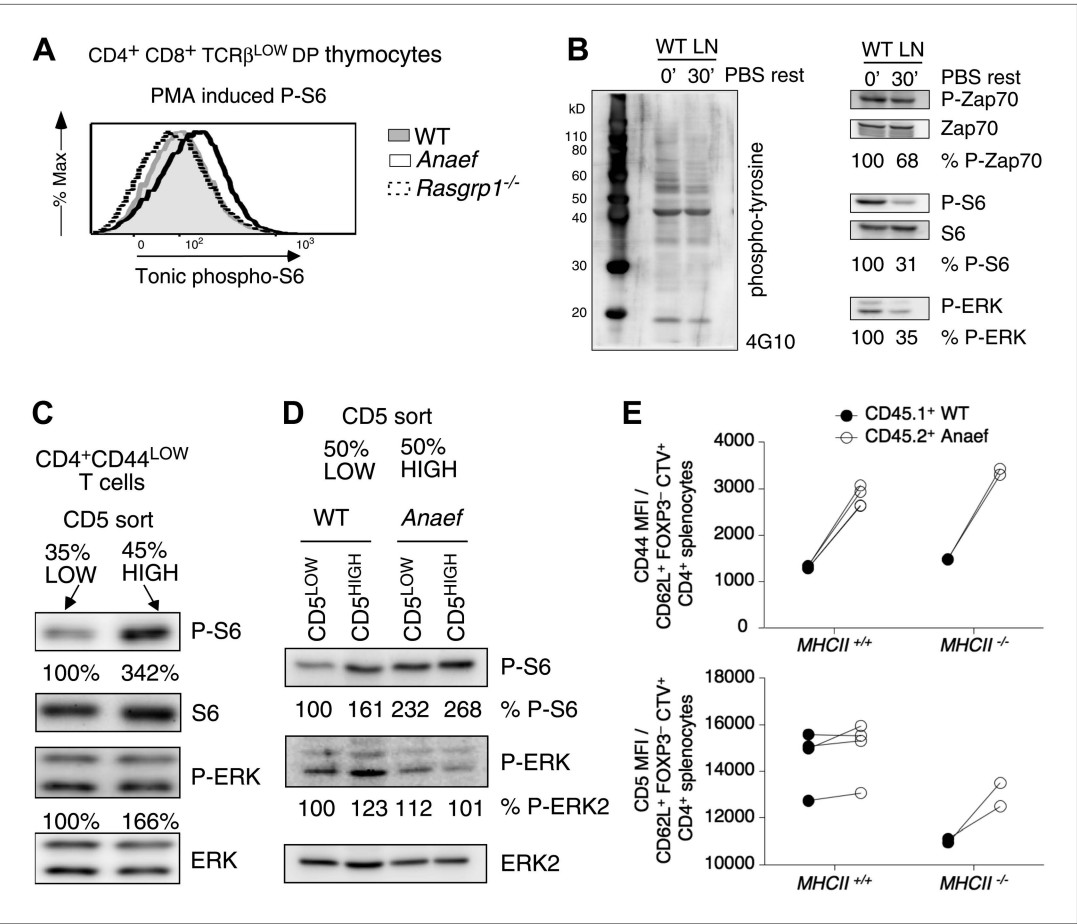

**Figure 10**. Increased tonic mTOR-S6 signals in *Rasgrp1^Anaef^* thymocytes and T cells. (**A**) Tonic P-S6 levels in pre-selection DP thymocytes from wildtype, *Rasgrp1^Anaef^* and *Rasgrp1^−/−^* mice. Representative histograms are shown for five independent experiments with n = 9 for WT, n = 7 for *Rasgrp1^Anaef^* and n = 7 *Rasgrp1^−/−^* mice. (**B**) Total lymph node cells were analyzed for the indicated phospho-proteins either immediately after extraction and single cell suspension generation (0') or after a 30 min rest period in PBS at 37° (30'). (**C, D**) Western blot measurements of basal P-S6 and P-ERK in unstimulated naive CD4+CD44^lo^ T cells sorted into CD5^low^ and CD5^high^ subsets from wildtype and *Rasgrp1^Anaef^* mice. Equal loading was confirmed with specific blotting for ERK2, which can be done without stripping. Panels c-e are representative results of at least three independent experiments. (**E**) CD45.1 wild-type and *Rasgrp1^Anaef^* splenocytes were mixed, labeled with CellTrace Violet and adoptively transferred into CD45.2 wild-type or MHCII–deficient recipient mice (***Figure 10—figure supplement 1***). 48 hr later, donor-derived CD4+CD62L+FOXP3− cells in recipient spleens were resolved by flow cytometry and their CD44 (upper panel) and CD5 (lower panel) MFIs were plotted. Lines connect measurements from individual mice.

The following figure supplements are available for figure 10:

**Figure supplement 1**. Testing the role of MHCII recognition in constitutive CD44 overexpression by *Rasgrp1^Anaef^* naive CD4 T cells.

---

*Rasgrp1^Anaef^*'s effect on Ras/ERK signaling in vivo was much milder than expected from its effects in *in vitro* assays. In response to relatively strong in vitro stimuli, the *Rasgrp1^Anaef^* mutation results in impaired membrane recruitment and $T_{184}$ phosphorylation of Rasgrp1 as well as reduced activation of Ras-ERK, establishing that *Rasgrp1^Anaef^* is a hypomorphic (partial loss-of-function) allele under these conditions. By contrast, in vivo, *Rasgrp1^Anaef^* did not alter CD69 and TCRβ induction or positive selection of thymocytes, unlike the C-terminally deleted (*Rasgrp1^d/d^*) and knockout (*Rasgrp1^−/−^*) alleles which caused moderate and severe decreases in these processes, respectively. This may indicate that the cumulative Ras-ERK signals required for these events are sufficiently buffered or robust that they tolerate a modest reduction in Rasgrp1's Ras activating activity. Only when a TCR transgene was

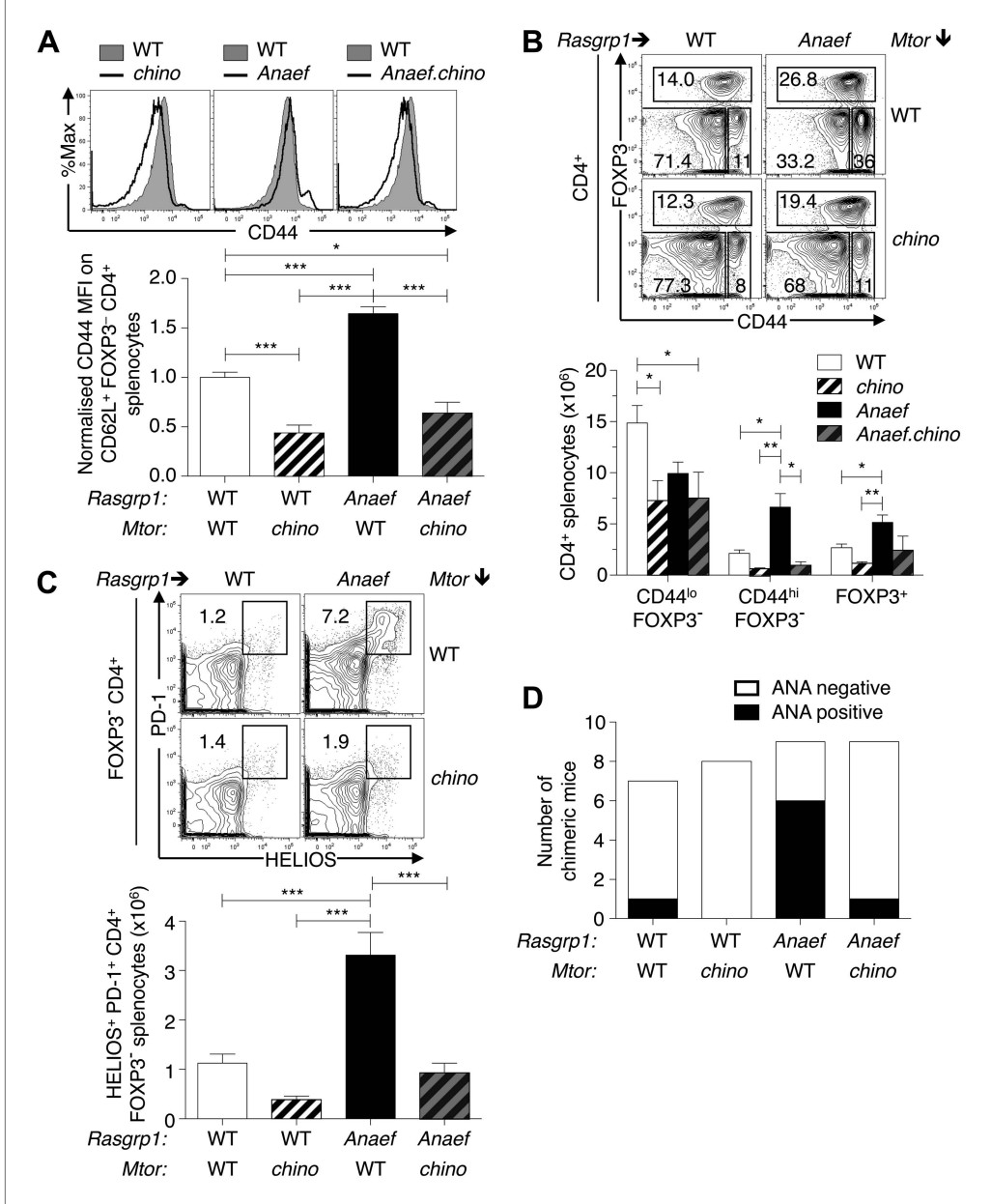

**Figure 11**. *Mtor* hypomorphic mutation corrects *Rasgrp1*[Anaef]–induced increase in naive T-cell CD44 expression and accumulation of CD44+Helios+PD-1+CD4+ cells and autoantibodies. (**A**) *B6.Rasgrp1*[Anaef] mice were intercrossed with *B6.Mtor*[chino] mice to generate the single and double-mutant mice. Representative CD44 histograms of CD62L+FOXP3−CD4+ splenocytes from these *chino*, *Anaef*, and *Anaef.chino* mutants were overlaid against wild-type cells and plotted. Below, CD44 MFI of mice was normalized against average CD44 MFI of wild-type mice across two independent experiments and graphed, with columns showing mean ± SEM. Significance indicated using a 1-way ANOVA and Tukey's post-test at n = 13 WT, 5 *chino*, 15 *Anaef*, and 4 *Anaef.chino*. *p<0.05, **p<0.01, ***p<0.001. (**B**) Representative CD44 vs FOXP3 plots for these four genotypes from (**A**), which display CD4+ splenocytes gated into naïve (CD44[lo] FOXP3−), activated/memory (CD44[hi] FOXP3−), and T-reg (FOXP3+) populations. Absolute numbers of these three subsets across all four genotypes is graphed below, with columns showing mean ± SEM. Significance indicated using a 1-way ANOVA and Tukey's post-test at n = 13 WT, 5 *chino*, 15 *Anaef*, and 4 *Anaef.chino*. *p<0.05, **p<0.01. (**C**) Bone marrow cells from sibling mice described in (**A**) and (**B**) were used to reconstitute irradiated *B6.SJL CD45*[1/1] mice, which were analyzed 28 weeks after irradiation. Plots show HELIOS vs PD-1 expression on FOXP3− CD4+ splenocytes, representative of both nonchimeric and chimeric mice. Absolute number of HELIOS+ PD-1+ FOXP3− CD4+ splenocytes per mouse is

*Figure 11. Continued on next page*

*Figure 11. Continued*

graphed below, with columns showing mean ± SEM. Significance indicated using a 1-way ANOVA and Tukey's post-test at n = 11 WT, 10 *chino*, 10 *Anaef*, 11 *Anaef.chino*. ***p<0.001. (**D**) Chimeric mice from (**C**) were bled 27 weeks after irradiation and the presence of homogeneous nuclear ANA in blood plasma was measured by immunofluorescence on HEp-2 cells and scored in a blinded manner.

prematurely expressed at higher than normal levels in DP thymocytes was there a measurable deficit in positive selection, and even under these conditions there was a small decrease in positive selection compared even to the subtle *Zap70murdock* mutation (*Siggs et al., 2007*). This is surprising given that the induction of these signals involves low affinity pMHC binding by the TCR, which might be expected to be particularly sensitive to small changes in Ras-ERK signal strength (*Kortum et al., 2013*).

CD44 expression on thymocytes is decreased in the complete absence of Rasgrp1 (*Figure 9*) (*Priatel et al., 2007*) and increased by oncogenic Ras (*Kindler et al., 2008*; *Zhang et al., 2009*; *Wang et al., 2011*) establishing that CD44 expression in thymocytes is positively regulated by Ras. T-cell CD44 expression is also sensitive to mTOR activity, being reduced by the partial loss-of-function *Mtorchino* (*Figure 8*) and *Mtortm1Lgm* (*Zhang et al., 2011*) alleles, and dramatically decreased in the absence of Rictor (*Delgoffe et al., 2011*), a binding partner of mTOR. *Rasgrp1Anaef* thymocytes and naïve CD4 T cells have increased CD44 and P-S6 expression, suggesting that the *Anaef* mutation increases either Rasgrp1/Ras/ERK signaling or PI-3-kinase/mTOR/S6 signaling, or both. Our recent biophysical studies revealed that Rasgrp1's EF hands keep the protein in an autoinhibited, dimeric state, and modeling indicates that calcium-binding to the EF domain would relieve autoinhibition (*Iwig et al., 2013*). Thus, Rasgrp1's EF hands play both stimulatory- and inhibitory-roles that may result in the EF2 substitution in *Rasgrp1Anaef* decreasing maintenance of the autoinhibited state in the absence of strong TCR stimuli and decreasing RasGRP1 activation during strong TCR stimulation. Evidence exists for multiple intersections between the RasGRP1/Ras/ERK/RSK and PI-3-kinase/mTOR/S6 pathways, including at the level of Ras with PI-3-kinase (*Castellano and Downward, 2010*) and at the level of RSK with S6 (*Salmond et al., 2009*). While the mechanism is currently unclear, the current evidence suggests that *Rasgrp1Anaef*'s gain-of-function in naïve T cells in the absence of strong TCR stimulation–and apparently in the absence of MHCII ligands for the TCR (*Figure 10E*)—activates S6-CD44 more than it activates ERK-CD69.

In T cells, the mTOR pathway is activated by strong TCR stimulation (*Gorentla et al., 2011*) and is required for efficient differentiation of naïve CD4 cells into effector cells (*Delgoffe et al., 2009*). T-cell-specific deletion of Tsc1, a negative regulator of mTOR, results in increased levels of P-S6 and an exuberant response to TCR stimulation in naïve T cells (*Yang et al., 2011*). Increased mTOR stimulation by Rasgrp1Anaef may allow self-antigens to activate some naïve CD4 cells, resulting in the gradual accumulation of activated CD62Llow CD44hi PD-1+ HELIOS+ T cells and antinuclear autoantibodies. Because accumulation of PD-1+ HELIOS+ T cells in *Rasgrp1Anaef* mice requires B cells (*Figure 7*), these T cells might require B cells as specialized APCs or they might require Fc receptor-dependent enhancement of antigen presentation by antibodies (*Silva et al., 2011*).

Given the huge number of missense variants in each person (*The 1000 Genomes Project Consortium, 2010*), patients with autoimmune diseases are more likely to have point mutations in various genes than complete loss of gene expression. This new *Rasgrp1Anaef* mouse model adds to an emerging category of animal models with point mutations in TCR signaling proteins, along with *Zap70skg* (*Sakaguchi et al., 2003*), *Zap70murdock* and *Zap70mrtless* hypomorphic alleles (*Siggs et al., 2007*), LATY136F mice (*Aguado et al., 2002*; *Sommers et al., 2002*), and *Card11unmodulated* mice (*Jun et al., 2003*), where a partial deficit in T cell signaling precipitates autoimmunity or allergy. *RASGRP1* splice variants have been documented for patients with SLE (*Yasuda et al., 2007*) and abnormal microRNA-driven downregulation of Rasgrp1 expression may play a role in aberrant DNA methylation in Lupus CD4+ T cells (*Pan et al., 2010*). In addition, *RASGRP1*-linked SNVs have been associated with autoimmune diabetes and thyroid disease (*Qu et al., 2009*; *Plagnol et al., 2011*). Of the 13 uncharacterized *RASGRP1* missense SNVs currently known, rs62621817 is of particular interest here since it causes a missense variation in RasGRP1's first EF hand, changing a conserved, negatively charged aspartic acid into a valine residue. We propose that *Rasgrp1Anaef* mice may provide a useful model system for further studies to help elucidate how *RASGRP1* variants contribute to autoimmune disease, and to help target future efforts to modulate this pathway pharmacologically.

## Materials and methods

### Mice

Mice were housed in pathogen-free conditions and experiments approved by either the Australian National University Animal Ethics and Experimentation Committee (Goodnow group, A2011/46) or the Institutional Animal Care and Use Committee of the University of California, San Francisco (Roose group, AN084051-01). C57BL/6 (B6), C57BL/6.SJL (CD45.1), B10Br, B10Br.CD45.1, B10Br TCR$^{3A9}$, *Cd79a$^{null}$* (also called *Cd79a$^{m1ANU}$*) and MHCII-deficient (*H2-Aa$^{tm1Blt}$*) mice were obtained from ANU Bioscience Services. The *Rasgrp1$^{Anaef}$* and *Mtor$^{chino}$* strains were established through ethylnitrosourea (ENU)-mediated mutagenesis of B6 mice at the Australian National University as previously described (*Randall et al., 2009*).

### Genetic mapping of the *Anaef* mutation

Affected Rasgrp1$^{Anaef}$ mice were crossed onto the CBA/J background to generate heterozygous F1 mice. F1 mice were intercrossed to yield mice homozygous for the *Anaef* mutation and carrying a mix of C57BL/6 and background CBA/J single nucleotide polymorphisms (SNPs). Genomic DNA samples isolated from both affected and unaffected mice were used as templates for SNP mapping at the Genomics Institute of the Novartis Research Foundation (San Diego, CA). SNP markers were spaced approximately every 3–5 Mbp throughout the genome. Once a defined interval was established, the Rasgrp1 encoding gene was sequenced from genomic DNA from both affected *Anaef* and WT mice. All exons were amplified by PCR with primers designed to include intronic RNA splice donor and acceptor sites. Exome enrichment using the SureSelect Mouse Exome kit (G7550A-001; Agilent, Santa Clara, CA), sequencing using the Illumina HiSeq 2000 (Illumina, San Diego, CA), and computational analysis to detect novel single-nucleotide variants were performed as described previously (*Andrews et al., 2012*).

### Genotyping

Roose lab *Anaef* mice were genotyped using MS-PCR. Primers were combined in a single reaction with Taq, Taq buffer and dNTPs (all New England BioLabs, Ipswich, MA). Goodnow lab *Anaef* mice were genotyped by APF Genomics Services following the manufacturer's instructions for Amplifluor PCR (SNP FAM/JOE; Millipore, Billerica, MA).

### Transfections and stable cell lines

Transfections and creation of stable cell lines was performed as previously described (*Roose et al., 2005*).

### Antinuclear antibody testing

Diluted mouse plasma was applied to HEp-2 slides (Inova, San Diego, CA). AlexaFluor488-conjugated goat anti-mouse IgG (Invitrogen, Carlsbad, CA) was added and slides mounted with fluorescence mounting medium (Dako Australia). Photos were taken using an Olympus IX71 microscope and WIB filter with 20 × lens and exposure time of 1/25 s.

### Flow cytometry

Suspensions of splenocytes (depleted of erythrocytes by brief osmotic lysis) or thymocytes were incubated with cocktails of anti-mouse antibodies specific for: CD44, CD4, CD45.1, CD45.2, CD5, CD62L, CD69, TCRβ, PD-1, TCR Vβ5, TCR Vβ8, TCR Vβ11, B220, CD95, GL-7, B220 (BD Pharmingen Franklin Lakes, NJ or BioLegend San Diego, CA) or CD8 (BD Pharmingen and UCSF Monoclonal Antibody Core, clone YTS169.4) conjugated to AlexaFluor700, APC-780 or APCCy7, PE-Cy7, APC, PerCPCy5.5, FITC, PE, Pacific Blue or biotin. Biotinylated antibodies were detected in another incubation step with streptavidin conjugated to Qdot605 (Invitrogen) or BV605 (BioLegend). Cells expressing the 3A9 TCR transgene were detected using the 1G12 (mouse IgG1) antibody (ATCC, Manassas, VA) followed by another incubation in anti-mouseIgG1 (A85.1). To detect intracellular proteins, cells were fixed and permeabilized using a Foxp3 staining kit (eBioscience, San Diego, CA), then labeled with antibodies specific for Foxp3 (FJK-16s; eBioscience), Helios (clone 22F6; Biolegend), IFNγ, IL-4, IL-2 or IL-17 (all BD Pharmingen). Flow cytometry data was acquired on a FACSort (Becton Dickinson) or an LSR Fortessa system and analyzed with FlowJo v8 (Treestar, Ashland, OR).

## Cell stimulations

Cells were stimulated using 25 ng/ml (HIGH), 5 ng/ml (MED), or 2 ng/ml (LOW) PMA (Calbiochem) with/without ionomycin (10 µM, Sigma, St. Louis, MO), or with 100 ng/ml PMA for Phospho-S6 assays. To mimic TCR engagement, cells were pre-labeled using anti-CD3 primary antibody (10 µg/ml, UCSF Monoclonal Antibody Core, clone 2C11) and crosslinking was achieved using goat anti-hamster antibody (10 µg/ml, Jackson ImmunoResearch). For intracellular cytokine detection by flow cytometry, splenocytes were stimulated in complete medium for 4 hr at 37°C with PMA (100 ng/ml; Sigma), ionomycin (500 ng/ml; Sigma) and GolgiStop (1/1000; BD), then labeled as described above.

## Flow cytometry for phosphorylated proteins

Procedure was performed as described in *Das et al., (2009)*. Cells were fixed using Cytofix Cell Fixative (BD Biosciences). Cells were permeabilized using 90% Methanol. Primary staining for phospho-Erk occurred using rabbit anti-mouse p-Erk antibody (#4377S; Cell Signaling) followed by staining with goat anti-Rabbit PE (Jackson ImmunoResearch, West Grove, PA). For analysis of tonic and PMA induced S6 phosphorylation by FACS, single-cell suspensions were prepared from thymus. Half of the cells were fixed in warm cytofix (BD Biosciences) immediately after harvesting and reserved for tonic signaling analysis. The remaining cells were then counted, and $10^6$ cells/sample were stimulated with PMA for 3 min, followed by fixation. Stimulated and unstimulated cells were then washed three times with cytoperm buffer (BD Biosciences) and incubated on ice in this buffer with a rabbit polyclonal antibody against phosphorylated S6 (cell signaling) for 45 min. The cells were then washed twice with cytoperm and incubated for an additional 45 min in cytoperm buffer containing an APC-conjugated goat anti-rabbit secondary antibody, as well as anti CD8-FITC, CD4, PE-Cy7 and TCRβ-PE. Cells were washed twice and analyzed in an LSR Fortessa system.

## Ras pulldown

Activation of Ras was analyzed using a RasGTP pulldown assay (Upstate Biotechnology, Lake Placid, NY) as previously described (*Roose et al., 2005*).

## Western blotting

Cells were lysed using Nonidet-P40 lysis buffer (1%) supplemented with protease and phosphatase inhibitors. Lysates were run on 10% acrylamide Bis-Tris gels and transferred onto PDVF filter (Millipore, Immobilon-P). Blots were probed for Rasgrp1 (M199; Santa Cruz, Dallas, TX), Alpha tubulin (Sigma), phospho-tyrosine (In house antibody prep, clone 4G10), phospho-Zap70 (Y493; Cell Signaling, Danvers, MA), phospho-PLCγ (Y783; Cell Signaling) and phospho-Lck (Y416; Cell Signaling), anti-ERK (pan-Erk) antibody (BD Transduction labs, clone 16/ERK). Rabbit anti-human Rasgrp1 (clone E80) was produced by Epitomics, Inc. (Burlingame, CA, USA). Mouse anti-Rasgrp1 p-T184 (clone JR-pT184RG1-4G7) was produced by AnaSpec (Fremont, CA, USA). Signal from primary antibodies detected using HRP conjugated secondary antibodies: Sheep anti-mouse HRP (GE Healthcare, Cleveland, OH) and goat anti-rabbit HRP (SouthernBiotech, Birmingham, AL). Blots were developed using Pierce ECL Western Blotting Substrate (ThermoScientific, Waltham, MA) and images recorded using a chemiluminescence imager (LAS-4000; Fuji).

## Bone marrow chimeras

Bone marrow was collected, and in some experiments, depleted of T cells and NK cells by magnetic labelling using biotinylated anti-TCRβ and anti-NK1.1 and streptavidin microbeads followed by passage through a MACS LD column (Miltenyi Biotec, Germany). Recipient mice were irradiated with X-rays (2 doses of 4.5 Gy given 4 hr apart) then injected intravenously (i.v.) with $2 \times 10^6$ bone marrow cells that were either from single or multiple donors as described in the text.

## Rapamycin treatment

Suboptimal doses of rapamycin (*Coenen et al., 2007*; *Araki et al., 2009*) were prepared on day 0 in sufficient quantity for all injections in one experiment. The appropriate rapamycin stock volume was diluted to the indicated concentrations in DMSO (15.4%), Cremaphor (15.4%) and water (69.2%), and aliquoted in six equal portions (one aliquot per injection) and frozen. Mice were injected on days 0, 1, 2, 3, 5 and 7, and then sacrificed on day 8. Thymocytes were harvested and analyzed as before.

## BrdU labelling

1 mg BrdU (BD) in PBS per mouse was injected i.p. to pulse label a cohort of dividing cells. Following surface staining of thymocytes as above, BrdU was detected following the BrdU Flow Kit (BD) protocol by fixing and permeabilizing cells with provided buffers, incubating for 1 hr at 37°C in DNase, then washing and staining with anti-BrdU antibody.

## CellTrace Violet (CTV) labelling

CTV labeling was done at room temperature as described (*Quah and Parish, 2010*) with slight modifications. Splenocytes suspended at $10^8$ cells/ml in RPMI containing HI-FCS (10% vol/vol) were transferred to the base of a fresh 15 ml conical tube. 1 μl of CTV (Life Technologies, Carlsbad, CA) stock solution (10 mM) per ml of cell suspension was placed on the dry wall of the tubes, then tubes were capped, inverted and briefly vortexed (final CTV concentration 10 μM). After 5 min incubation in the dark, 10 ml of 10%FCS/RPMI was added, then cells were sedimented by centrifugation before another wash in 10 ml of the same medium. Cells were then resuspended in PBS and passed through a 70 μm cell strainer (BD) before i.v. injection (200 μl per mouse).

## Acknowledgements

The authors would like to thank Drs Richard Glynne, Director of Genetics and Neglected Diseases at Novartis, for guiding the mapping project, Rich Lewis for discussion on calcium signals, and Michelle Hermiston for assistance with thymocyte FACS stainings. We thank Dr Jim Stone for sharing Rasgrp1 deficient mice and Dr Robert Barrington for sending these mice. We thank Debbie Howard, Nadine Barthel and Heather Domaschenz for expert technical assistance, and the animal services and genotyping teams at the Australian Phenomics Facility. We also thank the members of the Cyster, Goodnow, Vinuesa, and Roose labs and NHMRC Program members for helpful comments and suggestions.

## Additional information

### Funding

| Funder | Grant reference number | Author |
|---|---|---|
| Sandler Program in Basic Science | | Jeroen P Roose |
| National Institutes of Health | K01CA113367, ARRA supplement, R56-AI095292 | Jeroen P Roose |
| Wellcome Trust | 082030/B/07/Z | Christopher C Goodnow |
| Department of Innovation, Industry, Science, Research and Tertiary Education | | Edward M Bertram, Christopher C Goodnow |
| Clive and Vera Ramasciotti Foundation Grant | | Anselm Enders, Christopher C Goodnow |
| National Health and Medical Research Council | GNT1035858 | Anselm Enders |
| National Institutes of Health | 1R03AR062783-01A1 | Andre Limnander |
| National Institutes of Health | R01-AI74847 | Jason G Cyster |
| National Institutes of Health | R01 AI52127, U54 AI054523 | Edward M Bertram, Christopher C Goodnow |
| National Health and Medical Research Council | Program Grant and Australia Fellowship | Christopher C Goodnow |

The funders had no role in study design, data collection and interpretation, or the decision to submit the work for publication.

### Author contributions

SRD, KMC, DYH, KLR, CNJ, AL, Conception and design, Acquisition of data, Analysis and interpretation of data, Drafting or revising the article; DRM, NKP, AE, CR, BB, LAM, GS, EMB, MAF,

YS, TDA, BW, SWB, JRW, Acquisition of data, Analysis and interpretation of data; JGC, Conception and design, Analysis and interpretation of data; CCG, JPR, Conception and design, Analysis and interpretation of data, Drafting or revising the article

## Ethics

Animal experimentation: This study was performed in strict accordance with the recommendations in the Guide for the Care and Use of Laboratory Animals of the National Institutes of Health. All of the animals were handled according to approved institutional animal care and use committee (IACUC) protocols of the University of California San Francisco (UCSF; approval number AN084051-03) and of the Australian Phenomics Facility and the Australian National University (ANU; approval number A2011/46). These protocols were approved by the Committee on the Ethics of Animal Experiments of UCSF and ANU.

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
