## [Decision Letter]

Thank you for sending your work entitled “A missense variant of *Rasgrp1* increases tonic mTOR signals resulting in aberrant helper T cells and autoantibodies” for consideration at *eLife*. Your article has been favorably evaluated by a Senior editor and 3 reviewers, one of whom is a member of our Board of Reviewing Editors, and one of whom, Jonathan Powell, has agreed to reveal his identity.

The Reviewing editor and the other reviewers discussed their comments before we reached this decision, and the Reviewing editor has assembled the following comments to help you prepare a revised submission.

1) It seems likely that the *Rasgrp1*^*Anaef*^ mutation results in the loss of function of the molecules by a dominant negative effect and reduced expression of *Rasgrp1* (Figure 2 and Figure 4). However, the mutation enhanced CD44 expression, which is apparently opposite to the phenotype of *Rasgrp1*^*null*^ mutation. While the authors claim that a slight difference in mTOR-CD44 pathway would be responsible for the phenotypic difference, more precise analysis is required for addressing the difference. Similarly, tonic mTOR activation and reduced ERK signaling do not link to the observed effects, such as the increase of anti-nuclear antibodies and the accumulation of a CD44^hi^ T cells. In addition, with the finding that CD44, Helios, PD-1, CXCR5, Foxp3 and IFNg were upregulated in *Rasgrp1*^*Anaef*^ mice, can these phenotypes also be explained by the slight change of mTOR and the inhibition of ERK? Are there any common mechanisms for the regulation of these genes? Why does a fraction of CD4^+^ T cells remain CD44^lo^ ? Are those cells expressing a peculiar phenotype when compared to the CD44^hi^ CD4^+^ T cells?

2) Although the mutant mice showed significant reduction of the ERK signaling, the authors concluded that there were no abnormalities in CD4SP thymocytes, regulatory T cells, and clonal deletion. The authors need to examine whether their functions, such as suppressive activity of regulatory T cells, are normal.

3) In Figure 5, the reduced levels of P-T184 in the Rasgrp1^Anaef^ thymocytes may be due to the reduced levels of Rasgrp1^Anaef^ protein and to the poor stability of Rasgrp1^Anaef^ following TCR stimulation. To dismiss this possibility, the blot needs to be immune-blotted with an anti-Rasgrp1 antibody (as shown in Figure 2). In addition, the *Rasgrp1*^*Anaef*^ mutation could be more similar to the *Rasgrp1*^*null*^ mutation compared with the *Rasgrp1*^*+/-*^ mutation. To compare functional differences, it is recommended to include the *Rasgrp1*^*null*^ mutation in Figure 5.

4) In Figure 10, the data suggesting that “substantial” levels of P-S6 are found in freshly isolated lymph nodes need to be shown.

5) The conclusion that tonic TCR signal results in basal mTOR-S6 needs to be further substantiated since it is based on the aggregation of disparate data from the Germain laboratory and from the authors’ laboratory.

6) To identify the mutated gene, the authors narrowed down the region where the mutation located and picked up *Rasgrp1* because it was the only gene with an immune function within the located region. As some phenotypes were different between *Rasgap1*^*null*^ and *Rasgap1*^*Anaef*^ mutation, is there a possibility that another mutation locating on the adjacent region is involved in the observed phenotypes? In this context, if the progeny is generated from *Rasgrp1*^*Anaef*^ mice that are ANA negative (Figure 1), are they ANA positive?

7) It might be interesting if the authors quickly review papers that have genetically deleted mTOR signaling in T cells to see if there might be differences in CD44 expression in those mice that might not have been emphasized in the original papers.

---

## [Author Response]

*1) It seems likely that the* Rasgrp1^Anaef^
*mutation results in the loss of function of the molecules by a dominant negative effect and reduced expression of Rasgrp1 (*Figure 2
*and*
Figure 4*). However, the mutation enhanced CD44 expression, which is apparently opposite to the phenotype of* Rasgrp1^null^
*mutation. While the authors claim that a slight difference in mTOR-CD44 pathway would be responsible for the phenotypic difference, more precise analysis is required for addressing the difference. Similarly, tonic mTOR activation and reduced ERK signaling do not link to the observed effects, such as the increase of anti-nuclear antibodies and the accumulation of a CD44*^*hi*^
*T cells. In addition, with the finding that CD44, Helios, PD-1, CXCR5, Foxp3 and IFNg were upregulated in* Rasgrp1^Anaef^
*mice, can these phenotypes also be explained by the slight change of mTOR and the inhibition of ERK? Are there any common mechanisms for the regulation of these genes? Why does a fraction of CD4*^*+*^
*T cells remain CD44*^*lo*^
*? Are those cells expressing a peculiar phenotype when compared to the CD44*^*hi*^
*CD4*^*+*^
*T cells*?

We apologize for not being clear about the various consequences of the Anaef mutation in the original manuscript. We have made a number of changes to clarify the evidence that the Anaef mutation has 2 distinct effects: (1) it increases naïve T-cell CD44 expression and (2) it drives accumulation of CD44^hi^ effector CD4^+^ T cells and autoantibodies.

To distinguish between effects (1) and (2) above, new or modified figure panels demonstrate that *Rasgrp1*^*Anaef*^ increases CD44 expression in naïve T cells even in heterozygous *Rasgrp1*^*Anaef/+*^ mice (new Figure 1), in the absence of B cells (new Figure 7) or in the absence of MHCII expression (new Figure 10), in the presence of large numbers of wildtype Treg cells in mixed chimeras (new Figure 9) and increased CD44 levels are detectable even at the DP stage of thymocyte development (new Figure 9 and Figure 9–figure supplement 7B). The opposite effects of the null and Anaef alleles on CD44 expression in CD4SP thymocytes suggest that Anaef is in fact a gain-of-function allele under steady-state conditions in vivo. Rasgrp1’s autoinhibitory homodimer conformation recently published in *eLife* (Iwig et al.) offers an explanation for this: in the absence of calcium, the second EF hand (mutated in Anaef) conceals the DAG-binding C1 domain on the opposing monomer. We now cite papers that demonstrate intersections between the Ras and mTOR signaling pathways and conclude that, although the mechanism is unclear, Rasgrp1^Anaef^’s gain-of-function activates the mTOR pathway more strongly than it activates the ERK pathway.

The immediacy of effect (1) contrasts with the gradual, age-dependent nature of effect (2), i.e., accumulation of CD44^hi^ CD4^+^ T cells (Figure 6). These 2 effects are also separable in that only effect (2) requires B cells (Figure 7 with new panel 7D). Importantly, both effects are rectified by the reduction-of-function *Mtor*^*chino*^ mutation (Figure 11 with new panel 11A), demonstrating a central role for mTOR.

Thanks to the constructive criticism we received, we believe that the changes underlined in our revised Title, Abstract, Introduction, Results, and Discussion help to explain our data more clearly, and offer testable hypotheses for the underlying mechanisms.

*2) Although the mutant mice showed significant reduction of the ERK signaling, the authors concluded that there were no abnormalities in CD4SP thymocytes, regulatory T cells, and clonal deletion. The authors need to examine whether their functions, such as suppressive activity of regulatory T cells, are normal*.

We have clarified in text in the Discussion that although we do not observe any deficit in frequency of Foxp3^+^ Tregs, we have not tested if these Tregs have normal suppressive activity. We do conclude that should any deficit in Treg suppressive activity exist, it does not account for the increased CD44 on naïve CD4s nor the increased formation of CD4 effectors, because those abnormalities are cell autonomous and not corrected in mixed chimeras where there are large numbers of wildtype Tregs. The following text has been added:

“A putative defect in regulatory T cells cannot explain the increased CD44 expression on naïve *Rasgrp1*^*Anaef*^ CD4^+^ cells, because in mixed chimeras bearing many wild-type Foxp3^+^ CD4 cells, CD44 expression was still increased on *Rasgrp1*^*Anaef*^ but not on co-resident wild-type CD62L^+^ FOXP3^–^ CD4^+^ splenocytes.”

*3) In*
Figure 5*, the reduced levels of P-T184 in the Rasgrp1*^*Anaef*^
*thymocytes may be due to the reduced levels of Rasgrp1*^*Anaef*^
*protein and to the poor stability of Rasgrp1*^*Anaef*^
*following TCR stimulation. To dismiss this possibility, the blot needs to be immune-blotted with an anti-Rasgrp1 antibody (as shown in*
Figure 2*). In addition, the* Rasgrp1^Anaef^
*mutation could be more similar to the* Rasgrp1^null^
*mutation compared with the* Rasgrp1^+/-^
*mutation. To compare functional differences, it is recommended to include the* Rasgrp1^null^
*mutation in*
Figure 5.

We agree with this point. Indeed the total Rasgrp1 expression levels that accompany Figure 5 (and 5B) were shown in supplemental figure S4c of the original manuscript, and these immunoblots demonstrate that the loss of T184 phosphorylation is not due to the decreased expression of total Rasgrp1. For the sake of clarity we have now eliminated that figure and included all the controls in Figure 5.

Regarding direct comparisons with Rasgrp1 null thymocytes, TCR-induced ERK activation is severely impaired in Rasgrp1 null thymocytes and this has been published (e.g., Dower et al., *Nature Immunology*). Since Rasgrp1 deficiency results in a severe block in thymocyte development, Rasgrp1 null thymi consist almost entirely of CD4CD8 double positive thymocytes with low levels of TCR expression. There are virtually no CD4 or CD8 single positive cells that are more mature and express higher levels of surface TCR. Very differently, WT of Rasgrp1^Anaef^ thymi contain all normal thymocyte populations. Functional comparisons of TCR-induced PLCgamma and ERK activation in Rasgrp1^Anaef^ (or WT) thymocytes with the signals in Rasgrp1^null^ thymocytes are not very informative (and perhaps even confusing for the reader) since these comparisons involve not only different developmental thymocyte subsets but also have the caveat of comparing cells with different levels of TCR expression, which is known to affect the efficiency of TCR stimulation.

*4) In*
Figure 10*, the data suggesting that “substantial” levels of P-S6 are found in freshly isolated lymph nodes need to be shown*.

We thank the reviewers for this suggestion – these data have now been added as new Figure 10, which shows immunoblots of extracts from lymph node cells before or after a 30 minute PBS rest, and probing for total phospho-tyrosine, pZap70, pErk, and pS6. The blots clearly demonstrate that phosphorylated signaling molecules, including S6, are readily detectable in freshly isolated samples and that these levels decrease upon brief resting in PBS. We also moved a panel from the supplement to become new Figure 10, which shows tonic S6 signals in purified CD4 T cells.

*5) The conclusion that tonic TCR signal results in basal mTOR-S6 needs to be further substantiated since it is based on the aggregation of disparate data from the Germain laboratory and from the authors’ laboratory*.

We thank the reviewers for this important query. We performed an experiment to test the role of self- peptide/MHCII recognition, and found that the elevated CD44 levels on naïve CD4 T cells from the Anaef mice is retained when cells are transferred into MHCII knockout mice These results are presented in new Figure 10. Therefore we inserted the following text into the Results:

“To examine if self-peptide/MHCII recognition plays a role in the increased CD44 expression on *Rasgrp1*^*Anaef*^ naïve CD4^+^ T cells, we adoptively transferred a mixture of wild-type and *Rasgrp1*^*Anaef*^ splenocytes into wild-type or MHCII(*H2-Aa*)–deficient recipient mice…”

*6) To identify the mutated gene, the authors narrowed down the region where the mutation located and picked up* Rasgrp1 *because it was an only gene with an immune function within the located region. As some phenotypes were different between* Rasgap1^null^
*and* Rasgap1^Anaef^
*mutation, is there a possibility that another mutation locating on the adjacent region is involved in the observed phenotypes? In this context, if the progeny is generated from* Rasgrp1^Anaef^
*mice tha are ANA negative (*Figure 1*), are they ANA positive*?

The fact that RasGRP1-deficient Jurkat T cells reconstituted with a Rasgrp1^Anaef^ allele and *Rasgrp1*^*Anaef*^ thymocytes demonstrated very similar Ras-ERK responses upon in vitro stimulation (Figures 4 and 5) already argued against another mutation causing the phenotype in *Rasgrp1*^*Anaef*^ mice. However, to be certain there was no other mutation closely linked to the *Rasgrp1*^*Anaef*^ mutation, whole-exome capture, sequencing and computational analysis were performed on DNA from an affected mouse. This identified the *Rasgrp1*^*Anaef*^ mutation as the only novel homozygous single-nucleotide variant on chromosome 2 between 104.9 Mb and 145.7 Mb, a wide interval in which each boundary was marked by two affected F2 mice that inherited recombined chromosomes (Figure 1—figure supplement 1). We added the following text:

“Whole-exome capture, sequencing and computational analysis of DNA from an affected mouse (4) identified this mutation as the only novel single-nucleotide variant within the interval of interest on chromosome 2 (data not shown).”

*7) It might be interesting if the authors quickly review papers that have genetically deleted mTOR signaling in T cells to see if there might be differences in CD44 expression in those mice that might not have been emphasized in the original papers*.

We thank the reviewers for this suggestion and have added further details of the connection between CD44 levels and mTOR.

“This conclusion is reinforced by supplementary data from two studies: mice with a *neo*-insertion in an *Mtor* intron that decreases *Mtor* mRNA approximately 70% show a similar selective lowering of CD44 on CD4^+^ T cells (74) and CD44 levels are also reduced on naïve, CD62L-positive T cells that are deficient for the mTOR activator Rheb or deficient for the mTOR-interacting protein Rictor (16)…”